# Attentional modulation of neuronal variability in circuit models of cortex

**Tatjana Kanashiro[1,2,3], Gabriel Koch Ocker[2,3,4], Marlene R Cohen[3,5], Brent Doiron[2,3]***

[1]Program for Neural Computation, Carnegie Mellon University and University of Pittsburgh, Pittsburgh, United States; [2]Department of Mathematics, University of Pittsburgh, Pittsburgh, United States; [3]Center for the Neural Basis of Cognition, Pittsburgh, United States; [4]Allen Institute for Brain Science, Seattle, United States; [5]Department of Neuroscience, University of Pittsburgh, Pittsburgh, United States

**Abstract** The circuit mechanisms behind shared neural variability (noise correlation) and its dependence on neural state are poorly understood. Visual attention is well-suited to constrain cortical models of response variability because attention both increases firing rates and their stimulus sensitivity, as well as decreases noise correlations. We provide a novel analysis of population recordings in rhesus primate visual area V4 showing that a single biophysical mechanism may underlie these diverse neural correlates of attention. We explore model cortical networks where top-down mediated increases in excitability, distributed across excitatory and inhibitory targets, capture the key neuronal correlates of attention. Our models predict that top-down signals primarily affect inhibitory neurons, whereas excitatory neurons are more sensitive to stimulus specific bottom-up inputs. Accounting for trial variability in models of state dependent modulation of neuronal activity is a critical step in building a mechanistic theory of neuronal cognition.

**\*For correspondence:** bdoiron@pitt.edu

## Introduction

The behavioral state of the brain exerts a powerful influence on the cortical responses. For example, electrophysiological recordings from both rodents and primates show that the level of wakefulness (*Steriade et al., 1993*), active sensory exploration (*Crochet et al., 2011*), and attentional focus (*Treue, 2001*; *Reynolds and Chelazzi, 2004*; *Gilbert and Sigman, 2007*; *Moore and Zirnsak, 2017*) all modulate synaptic and spiking activity. Despite the diversity of behavioral contexts, in all of these cases an overall elevation and desynchronization of cortical activity accompanies heightened states of processing (*Harris and Thiele, 2011*). Exploration of the neuronal mechanisms that underlly such state changes has primarily centered around how various neuromodulators shift the cellular and synaptic properties of cortical circuits (*Hasselmo, 1995*; *Lee and Dan, 2012*; *Noudoost and Moore, 2011*; *Moore and Zirnsak, 2017*) However, a coherent theory linking the modulation of cortical circuits to an active desynchronization of population activity is lacking. In this study we provide a circuit-based theory for the known attention-guided modulations of neuronal activity in the visual cortex of primates performing a stimulus change detection task.

The investigation of the neuronal correlates of attention has a rich history. Attention increases the firing rates of neurons engaged in feature- and spatial-based processing tasks (*McAdams and Maunsell, 2000*; *Reynolds et al., 1999*). Attentional modulation of the stimulus-response sensitivity (gain) of firing rates is more complicated, often depending on stimulus specifics such as the size and contrast of a visual image (*Williford and Maunsell, 2006*; *Reynolds and Heeger, 2009*; *Sanayei et al., 2015*). In recent years there has been increased focus on how brain states affect trial-to-trial spiking variability (*Crochet et al., 2011*; *Lin et al., 2015*; *Doiron et al., 2016*; *Stringer et al.,*

**eLife digest** The world around us is complex and our brains need to navigate this complexity. We must focus on relevant inputs from our senses – such as the bus we need to catch – while ignoring distractions – such as the eye-catching displays in the shop windows we pass on the same street. Selective attention is a tool that enables us to filter complex sensory scenes and focus on whatever is most important at the time. But how does selective attention work?

Our sense of vision results from the activity of cells in a region of the brain called visual cortex. Paying attention to an object affects the activity of visual cortex in two ways. First, it causes the average activity of the brain cells in the visual cortex that respond to that object to increase. Second, it reduces spontaneous moment-to-moment fluctuations in the activity of those brain cells, known as noise. Both of these effects make it easier for the brain to process the object in question.

Kanashiro et al. set out to build a mathematical model of visual cortex that captures these two components of selective attention. The cortex contains two types of brain cells: excitatory neurons, which activate other cells, and inhibitory neurons, which suppress other cells. Experiments suggest that excitatory neurons contribute to the flow of activity within the cortex, whereas inhibitory neurons help cancel out noise. The new mathematical model predicts that paying attention affects inhibitory neurons far more than excitatory ones. According to the model, selective attention works mainly by reducing the noise that would otherwise distort the activity of visual cortex.

The next step is to test this prediction directly. This will require measuring the activity of the inhibitory neurons in an animal performing a selective attention task. Such experiments, which should be achievable using existing technologies, will allow scientists to confirm or disprove the current model, and to dissect the mechanisms that underlie visual attention.

*2016*). In particular, attention decreases the shared variability (noise correlations) of the firing rates from pairs of neurons (*Cohen and Maunsell, 2009*; *Mitchell et al., 2009*; *Cohen and Maunsell, 2011*; *Herrero et al., 2013*; *Ruff and Cohen, 2014*; *Engel et al., 2016*). The combination of a reduction in noise correlations and an increase in response gain has potentially important functional consequences through an improved population code (*Cohen and Maunsell, 2009*; *Rabinowitz et al., 2015*). In total, there is an emerging picture of the impact of attention on the trial-averaged and trial-variable spiking dynamics of cortical populations.

Phenomenological models of attentional modulation have been popular (*Reynolds and Heeger, 2009*; *Navalpakkam and Itti, 2005*; *Gilbert and Sigman, 2007*; *Ecker et al., 2016*); however, such analyses cannot provide insight into the circuit mechanics of attentional modulation. Biophysical models of attention circuits are difficult to constrain, due in large part to the diversity of mechanisms which control the firing rate and response gain of neurons (*Silver, 2010*; *Sutherland et al., 2009*). Nonetheless, several circuit models for attentional modulation have been proposed (*Ardid et al., 2007*; *Deco and Thiele, 2011*; *Buia and Tiesinga, 2008*), but analysis has been mostly confined to trial-averaged responses. Taking inspiration from these studies, mechanistic models of attentional modulation can be broadly grouped along two hypotheses. First, the circuit mechanisms that control trial-averaged responses may be distinct from those that modulate neuronal variability. This hypothesis has support from experiments in primate V1 showing that N-methyl-D-aspartate receptors have no impact on top-down attentional modulation of firing rates, yet have a strong influence of attentional control of noise correlations (*Herrero et al., 2013*). A second hypothesis is that the modulations of firing rates and noise correlations are reflections of a single biophysical mechanism. Support for this comes from pairs of V4 neurons that each show strong attentional modulation of firing rates, also show a strong attention mediated reductions in noise correlation (*Cohen and Maunsell, 2011*). In this study we provide novel analysis of the covariability of V4 population activity engaged in an attention-guided detection task (*Cohen and Maunsell, 2009*) that is consistent with the second hypothesis. Specifically, the modulation of spike count covariance between unattended and attended states has the same dimensionality as the firing rate modulation.

We use the results from our dimensionality analysis to show that an excitatory-inhibitory recurrent circuit model subject to global fluctuations is sufficient to capture both the increase in firing rate and

response gain as well as population-wide decrease of noise correlations. Our model makes two predictions regarding neuronal modulation: (1) that attentional modulation favors inhibitory neurons, and (2) that stimulus drive favors excitatory neurons. Finally, we show that our model predicts increased informational content in the excitatory population, which would result in improved readout by potential downstream targets. In total, our study provides a simple, parsimonious, and biologically motivated model of attentional modulation in cortical networks.

## Results

### Attention decreases noise correlations primarily by decreasing covariance

Two rhesus monkeys (*Macaca mulatta*) with microelectrode arrays implanted bilaterally in V4 were trained in an orientation change detection task (*Figure 1a*; see Materials and methods: Data preparation). A display with oriented Gabor gratings on the left and right flashed on and off. The monkey was cued to attend to either the left or right grating before each block of trials, while keeping fixation on a point between the two gratings. After a random number of presentations, one of the gratings changed orientation. The monkey then had to saccade to that side to obtain a reward. The behavioral task and data collection have been previously reported (*Cohen and Maunsell, 2009*).

A neuron is considered to be in an 'attended state' when the attended stimulus is in the hemifield containing that neuron's receptive field (contralateral hemifield), and in an 'unattended state' when it is in the other (ipsilateral) hemifield. The trial-averaged firing rates from both attended and

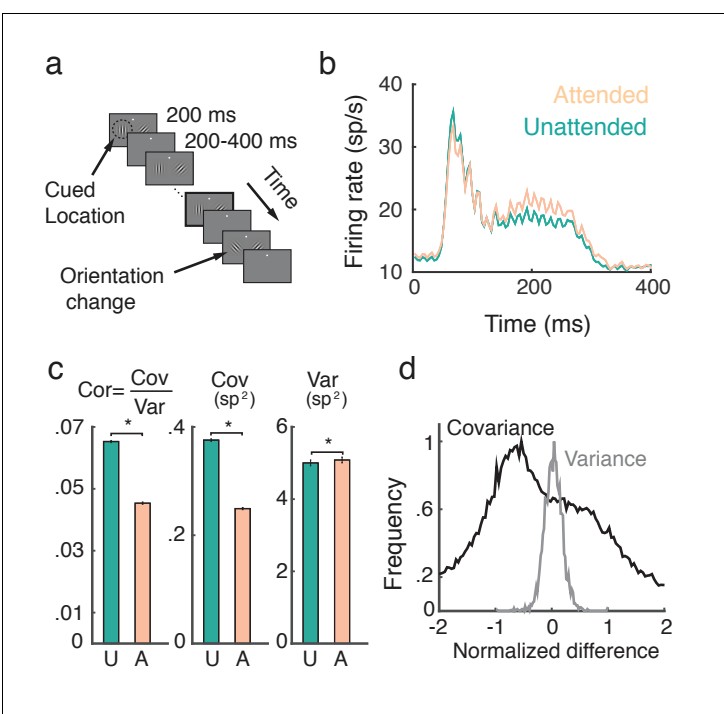

**Figure 1.** Attention increases firing rates and decreases trial-to-trial covariability of population responses. (**a**) Overview of orientation-change detection task; see (*Cohen and Maunsell, 2009*) for a full description. (**b**) Firing rates of neurons in the unattended (turquoise) and attended (orange) states, averaged over 3170 units. The slight oscillation in the firing rate was due to the monitor refresh rate. (**c**) Attention significantly decreased the spike count correlation and covariance and slightly increased variance. Error bars provide the SEM. (**d**) Histograms of changes in covariance for each unit pair (black) and variance for each unit (gray). In each case we consider the relative change $[X^A - X^U]/\max(X^A, X^U)$, where $X$ is either $\mathrm{Cov}(n_i, n_j)$ or $\mathrm{Var}(n_i)$. Data was collected from two monkeys over 21 and 16 recording sessions respectively. Signals were analyzed over a 200 ms interval, starting 60 ms after stimulus onset.

unattended neurons displayed a brief transient rise ($\sim 100$ ms after stimulus onset), and eventually settled to an elevated sustained rate before the trial concluded (**Figure 1b**). During the sustained period the mean firing rate of attended neurons (22.0 sp/s) was greater than that of unattended neurons (20.6 sp/s) ($t$ test, $P < 10^{-5}$).

A major finding of *Cohen and Maunsell (2009)* was that the pairwise trial-to-trial noise correlations of the neuronal responses decreased with attention (**Figure 1c**, left, mean unattended 0.065, mean attended 0.045, $t$ test, $P < 10^{-5}$). The noise correlation between neurons $i$ and $j$ is a normalized measure, $\rho_{ij} = \text{Cov}(n_i, n_j)/\sqrt{\text{Var}(n_i)\text{Var}(n_j)}$, where Cov and Var denote spike count covariance and variance respectively. Both spike count variance and covariance significantly change with attention ($\langle\text{Var}^U\rangle_{\text{trials}} = 5.02 \text{ spikes}^2$, $\langle\text{Var}^A\rangle_{\text{trials}} = 5.10 \text{ spikes}^2$, $t$ test, $P < 10^{-3}$, $\langle\text{Cov}^A\rangle_{\text{trials}} = 0.252$, $t$ test, $P < 10^{-5}$), but the decrease in covariance (34.0%) is much more pronounced than the increase in variance (1.61%; **Figure 1c**, middle and right). We therefore conclude that the attention mediated decrease in noise correlation is primarily due to decreased covariance.

To further validate this observation, we consider the distributions of pairwise changes in covariance (black) and variance (gray) with attention over the entire data set (**Figure 1d**). Covariance and variance are normalized by their respective maximal unattended or attended values (see Methods: Comparing change in covariance to change in variance). The change in covariance with attention is concentrated below zero with a large spread, whereas the change in variance is centered on zero with a narrower spread. Taken together these results suggest that to understand the mechanism by which noise correlations decrease it is necessary and sufficient to understand how spike count covariance decreases with attention.

## Attention is a low-rank modulation of noise covariance

A reasonable simplification of V4 neurons is that they receive a bottom-up stimulus alongside an attention-mediated top-down modulatory input. However, to properly model top-down attention we need to first understand the dimension of attentional modulation on the V4 circuit as a whole. Let $A_\phi : \phi^U \mapsto \phi^A$ denote the attentional modulation of measure $\phi$ from its value in the unattended state, $\phi^U$, to its value in the attended state, $\phi^A$. For example, the firing rate modulation $A_r$ can be written as $\mathbf{r}^A = A_r \circ \mathbf{r}^U$, where $\mathbf{r}^A$ is an $N \times 1$ vector of neural firing rates in the attended state, $\mathbf{r}^U$ denotes the firing rate vector in the unattended state, $A_r$ is a vector the same size as $\mathbf{r}$, and $\circ$ denotes elementwise multiplication. In this case, the entries $a_i$ of $A_r$ are the ratios of the firing rates: $a_i = r_i^A/r_i^U$ (**Figure 2a**).

A less trivial aspect of attentional modulation is the modulation of covariance matrices:

$$\mathbf{C}^A = A_C \circ \mathbf{C}^U. \tag{1}$$

Here $\mathbf{C^A}$ is the attended spike count covariance matrix, $\mathbf{C^U}$ the unattended spike count covariance matrix, and $A_C$ is a matrix the same size as $\mathbf{C^U}$, consisting of entries $g_{ij}$, which we will call *covariance gains*. Unlike firing rates, the transformation matrix $A_C$ can be of varying rank. On the one hand $A_C$ could be constructed from the ratios of the individual elements: $g_{ij} = c_{ij}^A/c_{ij}^U$, with each pair of neurons $(i,j)$ receiving an individualized attentional modulation $g_{ij}$ of their shared variability (**Figure 2b**, left). Under this modulation $A_C$ is a rank $N$ matrix. A rank $N$ $A_C$ will always perfectly (and trivially) capture the matrix mapping in **Equation (1)**. However, it is difficult to conceive of a top-down circuit mechanism that would allow attention to modulate each pair individually. On the other hand, $g_{ij}$ could depend not on the specific pair $(i,j)$, but on the individual neurons of the pairing: $g_{ij} = g_i g_j$ (**Figure 2b**, right). In this case, only $N$ values are needed to characterize $A_C : A_C = \mathbf{g}\mathbf{g}^T$, where $\mathbf{g}$ is a $N \times 1$ column vector, meaning $A_C$ has rank of 1. This is a more parsimonious and biophysically plausible scenario for attentional modulation, since in this case the covariance gain $g_{ij}$ of neurons $i$ and $j$ is simply emergent from the attentional modulation of the individual neurons. To test whether $A_C$ is low rank we analyzed the V4 population recordings during the visual attention task (**Figure 1**), specifically measuring $A_C$ under the assumption that $A_C$ is rank 1:

$$\mathbf{C}^A = \mathbf{g}\mathbf{g}^T \circ \mathbf{C}^U. \tag{2}$$

*Equation (2)* is a system of $N(N-1)/2$ equations of the form $c_{ij}^A = g_i g_j c_{ij}^U$ in $N$ unknowns $\mathbf{g} =$

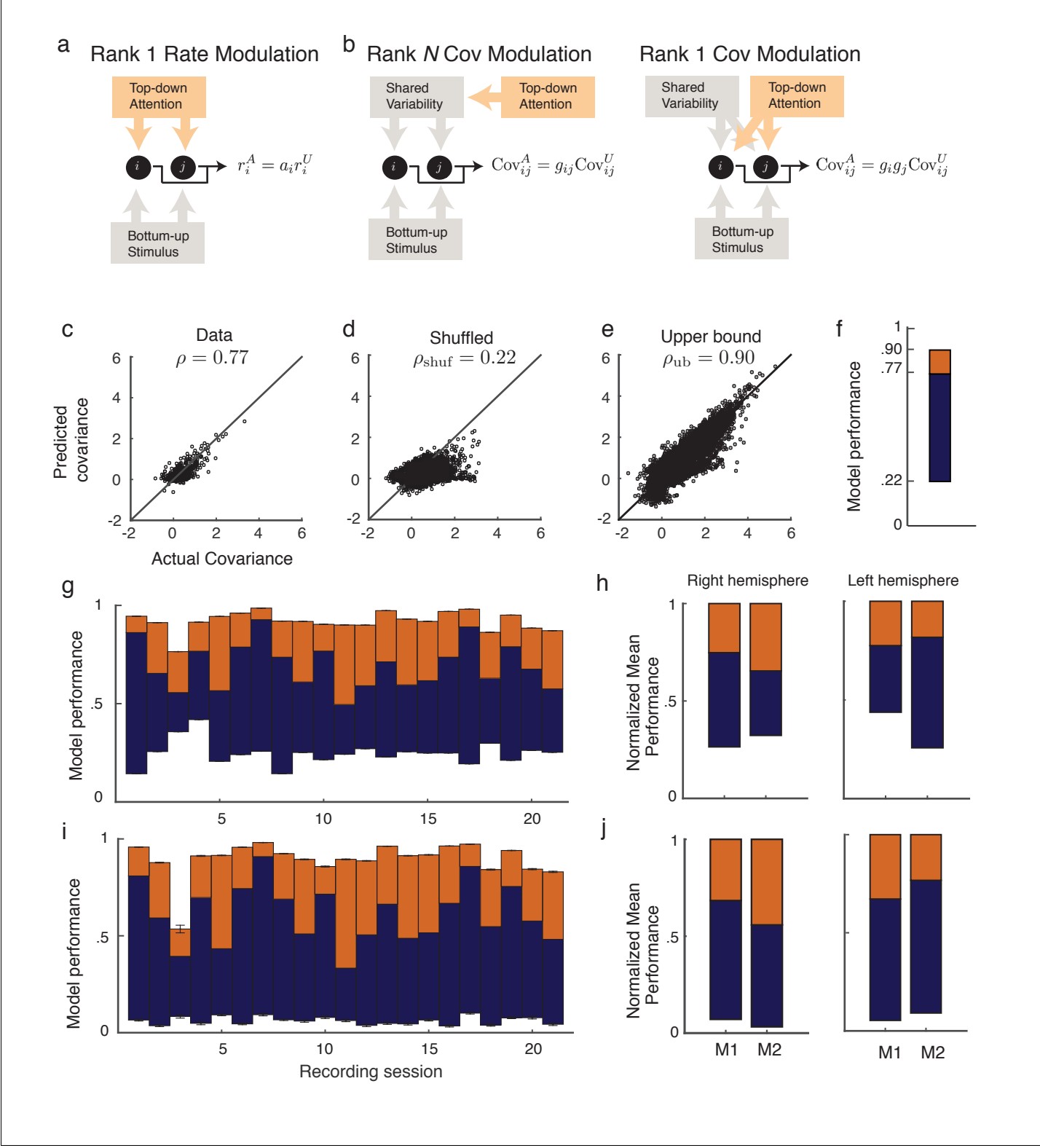

**Figure 2.** Rank one structure of attentional modulation of spike count covariance. (**a**) Attentional modulation of firing rate. Firing rates of neurons $i$ and $j$ (black circles are modulated by bottom-up stimulus and top-down attention. (**b**) Two possible models of attentional modulation of covariance. Left: High-rank covariance modulation, in which attention modulates the shared variability of each pair of neurons. Right: Low-rank covariance modulation, in which attention modulates each neuron individually rather than in a pairwise manner. (**c–e**) The measured covariance values plotted against those predicted by the rank-1 model for data collected in one recording session, for c, the actual data ($\rho = 0.77$), d, shuffled data ($\rho_{\mathrm{shuf}} = 0.22$, 100 shuffles),

*Figure 2 continued on next page*

*Figure 2 continued*

and (e) artificial upper-bound data ($\rho_{ub} = 0.90$, 10 realizations of the upper bound model). (f) Synthesis of c-e in a bar plot. The orange area represents the loss of model performance compared to the upper bound model, and the blue area represents the increase in model performance compared to model applied to shuffled data. (g) Rank-1 model performance reported for 21 recording sessions from one monkey. Each bar represents one recording session. Recordings from a mean of $N = 53.5$ units in the right-hemisphere were analyzed, with maximum and minimum $N$ of 80 and 35, respectively. Error bars denote standard error of the mean. (h) Mean normalized performance (relative to $\rho_{ub}$) for both hemispheres of two monkeys (M1 and M2). (i), Analysis as in (g), using leave-one-out cross-validation to test the predictive power of the model. (j) Mean normalized performance of the cross-validated data.

$[g_1, \ldots g_N]^T$ (we only consider $i \neq j$ to exclude variance modulation from our analysis). For $N > 3$ this is an overdetermined system, and we solve for g using a nonlinear equation solver. Let $\hat{g}$ be the optimal solution obtained by the solver (measured as a minimization of the $L^2$-norm of the error; see Methods: objfxn). Then $\hat{C}^A := \hat{g}\hat{g}^T \circ C^U$ provides an approximation to the attended covariance matrix. In an example data set from a single recording session with $N = 39$ units, the correlation coefficient $\rho$ of the actual attended covariance values from $\mathbf{C^A}$ versus the approximated attended covariance values from $\hat{C}^A$ was 0.77 (*Figure 2c*). A shuffled $\mathbf{C^A}$ matrix provides a reasonable null model, and the example data set produces the lower bound correlation $\rho_{shuf} = 0.22$ (*Figure 2d*; see Materials and methods: Shuffled covariance matrices). Finally, a Poisson model that perfectly decomposes as *Equation (2)*, yet sampled with the same number of trials as in the experiment, gives an upper bound for the rank one structure, the example data yields $\rho_{ub} = 0.90$ (*Figure 2e*; see Materials and methods: Upper bound covariance matrices). In total, the combination of $\rho$, $\rho_{shuf}$, and $\rho_{ub}$ (*Figure 2f*) suggests that the rank one model of attention modulation of covariance $A_C$ is well justified.

We applied this analysis to 21 recording sessions from the right hemisphere of one monkey (*Figure 2g*). For most of the recording sessions $\rho$ is closer to $\rho_{ub}$ than $\rho_{shuf}$. The averaged performance of all sessions for both hemispheres of two monkeys generally agreed with this trend (*Figure 2h*). We normalized $\rho$ and $\rho_{shuf}$ by $\rho_{ub}$ for each session to better compare different sessions that were subject to day-to-day variations outside of the experimenter's control, such as the task performance or the internal state of the monkey. To further validate our model we show the distribution of $g_i$s computed from the entire data set (*Figure 3a*). The majority of $g_i$ values are less than one, consistent with $\langle \mathrm{Cov}^A \rangle_{\mathrm{trials}} < \langle \mathrm{Cov}^U \rangle_{\mathrm{trials}}$ (*Figure 1c*). Further, there was little relation between the attentional modulation of firing rates, measured by $r_i^A / r_i^U$, and the attentional modulation of

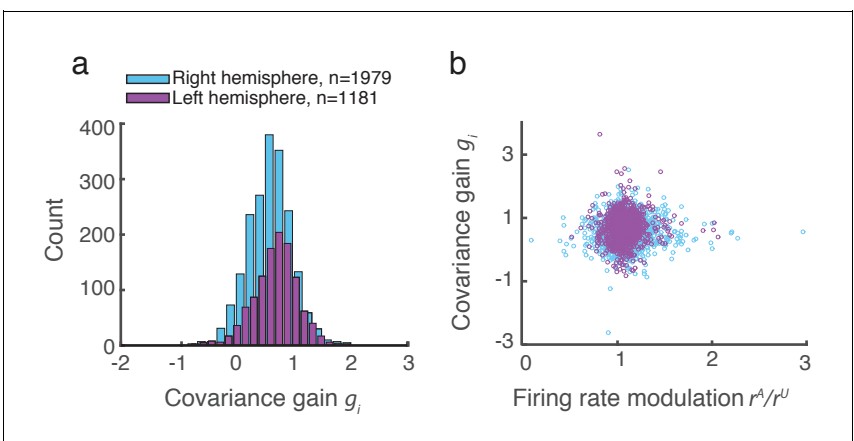

**Figure 3.** Covariance gain shows the attenuation of population-wide fluctuations with attention. (a) Distribution of covariance gains $g_i$ computed from the entire data set. (b) The relation between covariance $g_i$ and the attention mediated modulation of firing rates $r_i^A / r_i^U$. The correlation coefficients between the data sets were 0.036 and 0.051 for the right and left hemispheres, respectively.

covariance through $g_i$ (**Figure 3b**). This indicates that the circuit modulation of firing rates and covariance are not trivially related to one another (**Doiron et al., 2016**).

We additionally tested the validity of our model in **Equation (2)** with a leave-one-out cross-validation analysis (see Materials and methods: Leave-one-out cross-validation). We accurately predicted an omitted covariance $C_{ij}^A$ (**Figure 2i and j**), consistent with our original analysis (**Figure 2g and h**). The individual session-by-session performance values for both the standard and leave-one-out set-ups are provided (Appendix: Model performance for all monkeys and hemispheres).

Finally, we investigated to what extent the actual value of the covariance gain $g_i$ of neuron $i$ depends on the population of neurons in which it was computed. We solved the system of equations $C_{ij}^A = g_i g_j C_{ij}^U$ using covariance matrices computed from recordings from distinct sets of neurons, overlapping only by neuron $i$. This gives two estimates of $g_i$, that nevertheless agreed largely with one another (Appendix: Low-dimensional modulation is intrinsic to neurons). This supported the hypothesis that covariance gain $g_i$ is an intrinsic property of neuron $i$.

The standard and cross-validation tests verify that the low-rank model of attentional modulation defined in **Equation (2)** explains between 66 and $82\%$ (standard), or 56 and $77\%$ (cross-validation) of the data. Taking this to be a positive result, we conclude that the covariance gain modulation depends largely on the modulation of individual neurons.

## Network requirements for attentional modulation

Having described attentional modulation statistically our next goal is to develop a circuit model to understand the process mechanistically. Consider a network of $N$ coupled neurons, and let the spike count from neuron $i$ on a given trial be $y_i$. The network output has the covariance matrix $\mathbf{C}$ with elements $c_{ij} = \text{Cov}(y_i, y_j)$. In this section we identify the minimal circuit elements so that the attentional mapping $A_{\mathbf{C}} : \mathbf{C}^U \mapsto \mathbf{C}^A$ satisfies the following two conditions (on average):

**C1:** $c_{ij}^A = g_i g_j c_{ij}^U$ ; attentional modulation of covariance is rank one (**Figure 2**).

**C2:** $g_i < 1$ ; spike count covariance decreases with attention (**Figure 1**).

What follows is only a sketch of our derivation (a complete treatment is given in Appendix: Network requirements for attentional modulation).

If inputs are weak then $y_i$ can be described by a linear perturbation about a background state (**Ginzburg and Sompolinsky, 1994**; **Doiron et al., 2004**; **Trousdale et al., 2012**):

$$y_i = y_{iB} + L_i \left( \sum_{k=1}^{N} J_{ik} y_k + \xi_i \right). \tag{3}$$

Here $y_{iB}$ is the background activity of neuron $i$, $J_{ik}$ is the coupling strength from neuron $k$ to $i$, and $L_i$ is the input-to-output gain of neuron $i$. In addition to internal coupling we assume a source of external fluctuations $\xi_i$ to neuron $i$. Here $y_i$, $y_{iB}$, and $\xi_i$ are random variables that vary across trials. The trial-averaged firing rate of neuron $i$ is $r_i = \langle y_i \rangle / T$ (where $\langle \cdot \rangle$ denotes averaging over trials of length $T$). The background state has variability $b_i = \text{Var}(y_{iB})$ which we assume to be independent across neurons, meaning the background network covariance is $\mathbf{B} = \text{diag}(b_i)$. Finally, the external fluctuations have covariance matrix $\mathbf{X}$ with element $x_{ij} = \text{Cov}(\xi_i, \xi_j)$.

Motivated by our analysis of population recordings (**Figure 2**) we study attentional modulations that target individual neurons. This amounts to considering only $A_r : r_i^U \mapsto r_i^A$ and $A_L : L_i^U \mapsto L_i^A$. Additionally, we assume that any model of attentional modulation must result in $r_i^A > r_i^U$ (**Figure 1b**). A widespread property of both cortical pyramidal cells and interneurons is that an increase of firing rate $r_i$ causes an increase of input-output gain $L$ (**Cardin et al., 2007**), thus we will also require $L^A > L^U$.

Spiking covariability in recurrent networks can be due to internal interactions (through $J_{ik}$) or external fluctuations (through $\xi_i$), or both (**Ocker et al., 2017**). Networks with unstructured connectivity have internally generated covariability that vanishes as $N$ grows. This is true if the connectivity is sparse (**van Vreeswijk and Sompolinsky, 1998**), or dense having weak synapses where $J_{ik} \sim 1/N$ (**Trousdale et al., 2012**) or strong synapses where $J_{ik} \sim 1/\sqrt{N}$ combined with a balance between excitation and inhibition (**Renart et al., 2010**; **Rosenbaum et al., 2017**). In these cases spiking covariability requires external fluctuations to be applied and subsequently filtered by the network. We follow this second scenario and choose $\mathbf{X}$ so as to provide external covariability to our network.

Recent analysis of cortical population recordings show that the shared spiking variability across the population can be well approximated by a rank one model of covariability (*Kelly et al., 2010*; *Ecker et al., 2014*; *Lin et al., 2015*; *Ecker et al., 2016*; *Rabinowitz et al., 2015*; *Whiteway and Butts, 2017*) (we remark that *Rabinowitz et al., 2015* analyzed the same data set that we have in *Figures 1* and *2*). Thus motivated we take the external fluctuations $\mathbf{X}$ to be rank one with $x_{ij} = x_i x_j$, reflecting a single source of global external variability $\xi$ with unit variance (neuron $i$ receives $\xi_i = x_i \xi$). Combining this assumption with the linear ansatz in *Equation (3)* yields:

$$\mathbf{C} \approx \left( (\mathbf{I} - \mathbf{K})^{-1} \mathbf{Lx} \right) \left( (\mathbf{I} - \mathbf{K})^{-1} \mathbf{Lx} \right)^T = \mathbf{cc}^T, \tag{4}$$

where matrix $\mathbf{K}$ has element $K_{ij} = L_i J_{ij}$ and $\mathbf{L} = \mathrm{diag}(L_i)$. We have also defined the vectors $\mathbf{x} = [x_1, \ldots, x_N]^T$ and $\mathbf{c} = [c_1, \ldots, c_N]^T$ with $c_i = ((\mathbf{I} - \mathbf{K})^{-1} \mathbf{Lx})_i$. In total, the output covariability $\mathbf{C}$ will simply inherit the rank of the input covariability $\mathbf{X}$. Attentional modulation affects $c_i$ through $\mathbf{K}$ and $\mathbf{L}$ and we easily satisfy condition **C1** with $g_i = c_i^A / c_i^U$.

What remains is to find constraints on $\mathbf{J}$ and the attentional modulation of $\mathbf{L}$ that satisfy condition **C2**. Let us consider the case where $c_i^U, c_i^A > 0$ so that condition **C2** is satisfied when $c_i^A - c_i^U < 0$. For the sake of mathematical simplicity let us separate the population into $qN$ excitatory neurons and $(1-q)N$ inhibitory neurons ($0 < q < 1$). Let all excitatory (inhibitory) neurons project with synaptic strength $J_E$ ($-J_I$), have gain $L_E$ ($L_I$), and receive the external inputs of strength $x_E$ ($x_I$). Finally, let the probability for all connections be $p$, and consider only weak connections ($J \propto 1/N$ and $N$ large) so that we can ignore the influence of polysynaptic paths in the network (*Pernice et al., 2011*; *Trousdale et al., 2012*). Then the attentional modulation of an excitatory neuron decomposes into:

$$c_E^A - c_E^U = \underbrace{\left( L_E^A - L_E^U \right) x_E}_{\text{direct external input}} + \underbrace{\left( L_E^A - L_E^U \right) qpNJ_E x_E}_{\substack{\text{external input filtered} \\ \text{through the excitatory population}}} - \underbrace{\left( L_I^A - L_I^U \right)(1-q)pNJ_I x_I}_{\substack{\text{external input filtered} \\ \text{through the inhibitory population}}}. \tag{5}$$

The first term is the direct transfer of the external fluctuations, and the second and third terms are indirect transfer of external fluctuations via the excitatory and inhibitory populations, respectively. Recall that $L^A - L^U > 0$, meaning that for $c_E^A - c_E^U < 0$ to be satisfied we require the third term to outweigh the combination of the first and second terms. In other words, the inhibitory population must experience a sizable attentional modulation. A similar cancelation of correlations by recurrent inhibition has been recently studied in a variety of cortical models (*Renart et al., 2010*; *Tetzlaff et al., 2012*; *Ly et al., 2012*; *Doiron et al., 2016*; *Rosenbaum et al., 2017*).

In the above we considered weak synaptic connections where $J_{ij} \sim 1/N$. Rather, if we scale $J_{ij} \sim 1/\sqrt{N}$, as would be the case for classical balanced networks (*van Vreeswijk and Sompolinsky, 1998*), then for very large $N$ the solution no longer depends upon the gain $L$. Finite $N$ or the inclusion of synaptic nonlinearities through short term plasticity (*Mongillo et al., 2012*) may be necessary to satisfy condition **C2** with large synapses. Furthermore, the large synaptic weights associated with $J_{ij} \sim 1/\sqrt{N}$ do not allows us to neglect polysynaptic paths, as is needed for *Equation (5)*. Extending our analysis to networks with balanced scaling will be the focus of future work.

In summary our analysis has identified two circuit features that allow recurrent networks to capture conditions **C1** and **C2** for attentional modulation. First, the network must be subject to a global source of external fluctuations that dominates network covariability (**C1**). Second, the network must have recurrent inhibitory connections that are subject to a large attentional modulation (**C2**).

## Mean field model of attention

We next apply the intuition gained in the preceding section to propose a cortical model that captures key neural correlates of attentional modulation. We model V4 as a recurrently coupled network of excitatory and inhibitory leaky integrate-and-fire model neurons (*Tetzlaff et al., 2012*; *Ledoux and Brunel, 2011*; *Trousdale et al., 2012*; *Doiron et al., 2004*) (*Figure 4a*). In addition to recurrent synaptic inputs, each neuron receives private and global sources of external fluctuating input (*Figure 4b*). The global noise is an attention-independent source of input correlation that the network filters and transforms into network-wide output spiking correlations (*Figure 4c*).

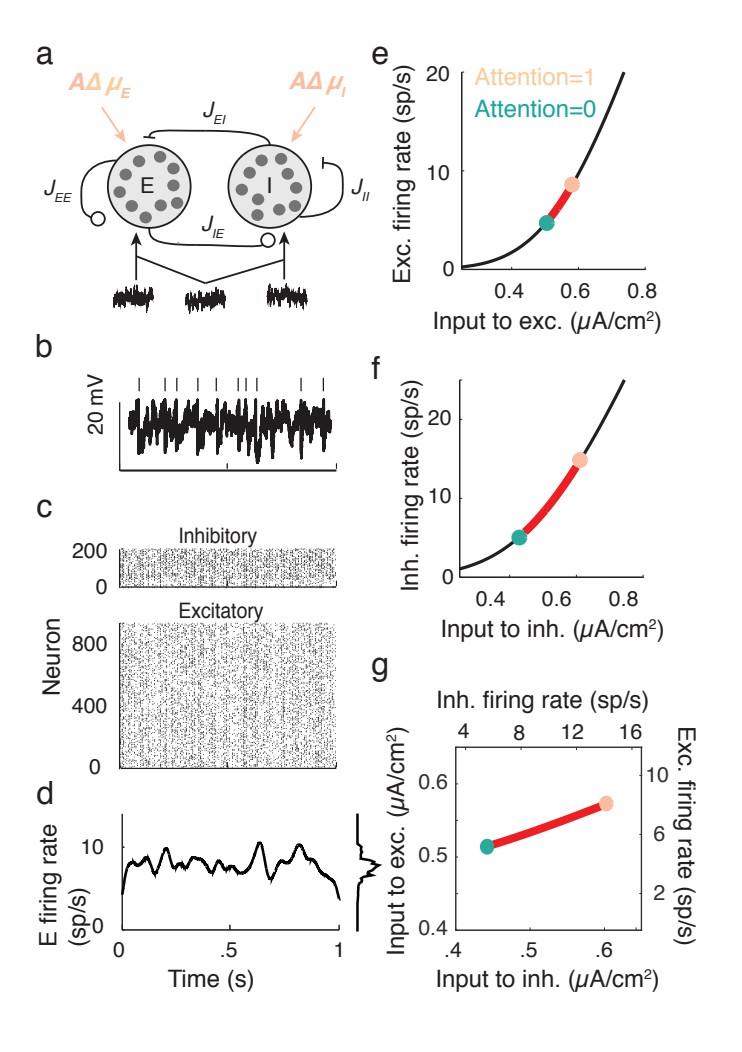

**Figure 4.** Excitatory-inhibitory network model. (**a**) Recurrent excitatory-inhibitory network subject to private and shared fluctuations as well as top-down attentional modulation. (**b**) Example voltage trace from a LIF model neuron in the network. Top tick marks denote spike times. (**c**) Spike time raster plot of the spiking activity from the model network. (**d**) Population-averaged firing rate $r_E(t)$ of the excitatory population. Left: frequency distribution of population-averaged firing rate. (**e**) Transfer function $f_E$ between the effective input and the firing rate for a model excitatory neuron. The red segment represents the attentional shift in effective input and hence firing rate. (**f**), Same as **e**, but for the inhibitory population. (**g**) Attention as a path through $(\bar{r}_E, \bar{r}_I)$ space, and equivalently through $(I_E^{\text{eff}}, I_I^{\text{eff}})$ space.

While the linear response theory introduced in *Equation (3)* is well suited to study large networks of integrate-and-fire neurons driven by weakly correlated inputs (*Tetzlaff et al., 2012*; *Ledoux and Brunel, 2011*; *Trousdale et al., 2012*; *Doiron et al., 2004*), the analysis offers little analytic insight. Instead, we consider the instantaneous activity across population $\alpha : r_a(t) = \frac{1}{N_\alpha}\sum_i y_{i\alpha}(t)$, where $y_{i\alpha}(t)$ is the spike train from neuron $i$ of population $\alpha$ and $N_\alpha$ is the population size ($\alpha = E$ or $I$). This approach reduces the model to just the two dynamic variables, the excitatory population rate $r_E(t)$ and the inhibitory population rate $r_I(t)$ ($r_E(t)$ is shown in *Figure 4d*). Despite this severe reduction the model retains the key ingredients for attentional modulation identified in the previous section – recurrent excitation and inhibition combined with a source of global fluctuations.

We take the population sizes to be large and consider a phenomenological dynamic mean field (*Tetzlaff et al., 2012*; *Ledoux and Brunel, 2011*) of the cortical network (see Materials and methods: Mean field model):

$$\tau_E \frac{dr_E}{dt} = -r_E + f_E(\mu_E + J_{EE}r_E - J_{EI}r_I + \sigma_E\xi(t)),$$

$$\tau_I \frac{dr_I}{dt} = -r_I + f_I(\mu_I + J_{IE}r_E - J_{II}r_I + \sigma_I\xi(t)).$$

(6)

The function $f_\alpha$ is the input-output transfer of population $\alpha$, taken to be the mean firing rate for a fixed input (*Figure 4e* for the $E$ population and *Figure 4f* for the $I$ population). The parameter $J_{\alpha\beta}$ is the coupling strength from population $\beta$ to population $\alpha$. Finally, $\mu_\alpha$ and $\sigma_\alpha$ are the respective strengths of the mean input and the global fluctuation $\xi(t)$ to population $\alpha$ (throughout $\xi(t)$ has a zero mean). To simplify our exposition we take symmetric coupling $J_{EE} = J_{IE} \equiv J_E$ and $J_{EI} = J_{II} \equiv J_I$ and symmetric timescales $\tau_E = \tau_I (= 1)$. We set the recurrent coupling so that the model has a stationary mean firing rate $(\bar{r}_E, \bar{r}_I)$, about which $\xi(t)$ induces fluctuations in $r_E(t)$ and $r_I(t)$.

Attention is modeled as a top-down influence on the static input: $\mu_\alpha = \mu_{\alpha B} + A\Delta\mu_\alpha$. Here $\mu_{\alpha B}$ is a background input, the parameter $A$ models attention with $A = 0$ denoting the unattended state and $A = 1$ the fully attended state, and $\Delta\mu_\alpha > 0$ is the increase in $\mu_\alpha$ due to attention. We note that the choice of representing the unattended state by $A = 0$ and the attended state by $A = 1$ is only due to convenience, and is not meant to make any statement about particular bounds on these states. In this model attention simply increases the excitability of all of the neurons in the network (*Figure 4a*). This modulation is consistent with the rank one structure of attentional modulation in the data (*Figure 2*), since $\mu_\alpha$ is a single neuron property. The attention-induced increase in $(\mu_E, \mu_I)$ causes an increase in the mean firing rates $(\bar{r}_E, \bar{r}_I)$ (red paths in *Figure 4e,f*), consistent with recordings from putative excitatory (*McAdams and Maunsell, 2000*; *Reynolds et al., 1999*) and inhibitory neurons (*Mitchell et al., 2007*) in visual area V4. Since $f_\alpha$ is a simple rising function then there is a unique mapping of an attentional path in $(\mu_E, \mu_I)$ space to a path in $(\bar{r}_E, \bar{r}_I)$ space (*Figure 4g*).

In total, our population model has the core features required to satisfy Conditions **C1** and **C2** of the previous section. We next use our mean field model to investigate how attentional paths in $(\bar{r}_E, \bar{r}_I)$ space affect population spiking variability.

## Attention modulates population variability

The global input $\xi(t)$ causes fluctuations about the network stationary state: $r_\alpha(t) = \bar{r}_\alpha + \delta r_\alpha(t)$. The fluctuations $\delta r_\alpha(t)$ are directly related to coordinated spiking activity in population $\alpha$. In particular, in the limit of large $N_\alpha$ we have that $V_E \equiv \text{Var}(r_E) \propto \langle\text{Cov}(y_i, y_j)\rangle$, where the expectation is over $(i, j)$ pairs in the spiking network. Thus, in our mean field network we require attentional modulation to decrease population variance $V_E$.

For sufficiently small $\sigma_\alpha$ the fluctuations $\delta r_E(t)$ and $\delta r_I(t)$ obey linearized mean field equations (see Materials and methods: Mean field model, *Equation (17)*). The linear system is readily analyzed and we obtain the population variance $V_E$ computed over long time windows (see Materials and methods: Computing $V_E$):

$$V_E = \left[\frac{L_E(J_I L_I(\sigma_E - \sigma_I) + \sigma_E)}{1 + J_I L_I - J_E L_E}\right]^2.$$

(7)

Here $L_\alpha \equiv f'_\alpha$ is the response gain of neurons in population $\alpha$. *Equation (7)* shows that $V_E$ depends directly on $L_\alpha$, and we recall that $L_\alpha$ changes with attention (the slope of $f_\alpha$ in *Figure 4e,f*). Thus, while the derivation of $V_E$ requires linear fluctuations about a steady state, attentional modulation samples the nonlinearity in the transfer $f_\alpha$ by changing the state about which we linearize. Any attention-mediated change in $V_E$ is not obvious since both $L_I^A > L_I^U$ and $L_E^A > L_E^U$, meaning that both the numerator and denominator in *Equation (7)* will change with attention.

We explore $V_E$ by sweeping over $(\bar{r}_E, \bar{r}_I)$ space (*Figure 5a*). When the network has high $\bar{r}_E$ and low $\bar{r}_I$ then $V_E$ is large, while $V_E$ is low for the opposite case of high $\bar{r}_I$ and low $\bar{r}_E$. Along our attention path $r_E$ increases while $V_E$ decreases (*Figure 5b*), satisfying our requirements for attentional modulation. The attention path that we highlight is just one potential path that reduces population variability, however all paths which reduce $V_E$ share a large attention-mediated recruitment of inhibition. If

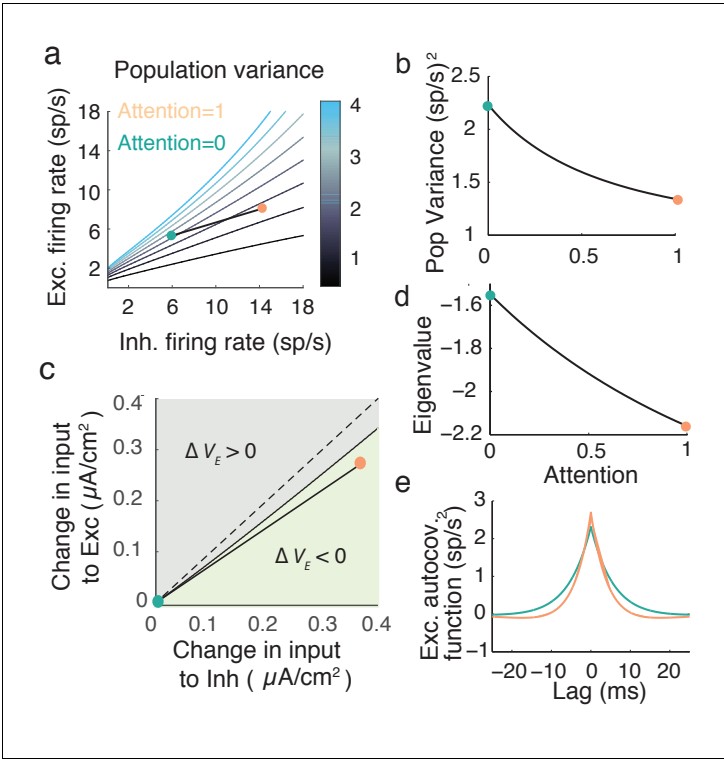

**Figure 5.** Mean field model shows an attention mediated decrease in population variance. (a) An attentional path in excitatory-inhibitory firing rate space for which the population variance decreases. Colored contours define iso-lines of population variance in increments of $10$ (sp/s)$^2$. The attentional path links the unattended state ($A = 0$; turquoise point) to the attended state ($A = 1$, orange point). (b) Variance values as a function of the attentional path defined in a. (c) The modulation from an unattended state (origin) to an attended state over the input space ($\Delta\mu_E, \Delta\mu_I$). Solid black line marks where $V_E$ remains unchanged, and the green region where $\Delta V_E = \text{Var}^A(r_E) - \text{Var}^U(r_E)$ is less than zero. (d) The eigenvalue ($\lambda$) along the attentional path. With increased attention it becomes more negative, indicating that the state ($\bar{r}_E, \bar{r}_I$) is more stable. e, Autocovariance function of the excitatory population rate $r_E(t)$ in the attended and unattended state (computed using **Equation (19)**).

we start with the unattended state (turquoise dot in **Figure 5c**) we can label all ($\Delta\mu_E > 0, \Delta\mu_I > 0$) points that have a smaller population variance than the unattended point (light green region in **Figure 5c**). These modulations all share that $\Delta\mu_I > \Delta\mu_E$ (**Figure 5c**, green region is below the $\Delta\mu_E = \Delta\mu_I$ line). While the absolute comparison between $\Delta\mu_E$ and $\Delta\mu_I$ may depend on model parameters, a robust necessary feature of top-down attentional modulation is that it must significantly recruit the inhibitory population. This observation is a major circuit prediction of our model.

An intuitive way to understand inhibition's role in the decrease in population variance is through the stability analysis of the mean field equations. The eigenvalues of the linearized system are $\lambda_1 = -1 - J_I L_I + J_E L_E < 0$ and $\lambda_2 = -1$ (see Materials and methods: Mean field model, **Equation (18)**). Note that the denominator of the population variance (**Equation 7**) equals the square of the eigenvalue product $\lambda_1 \lambda_2 = 1 + J_I L_I - J_E L_E$. The stability of the network activity is determined by $\lambda_1$; the more negative $\lambda_1$, the more stable the point ($\bar{r}_E, \bar{r}_I$), and the better the network dampens the perturbations about the point due to input fluctuations $\xi(t)$. The decrease of $\lambda_1$ along the example attention path is clear (**Figure 5d**), and overcomes the increase in the numerator of $V_E$ due to increases in $L_E$ and $L_I$. The enhanced damping is why $V_E$ decreases, explicitly seen in the steeper decline of the excitatory population autocovariance function in the attended compared to the unattended state (**Figure 5e**).

This enhanced stability due to recurrent inhibition is a reflection of inhibition canceling population variability provided by external fluctuations and recurrent excitation (**Renart et al., 2010**; **Tetzlaff et al., 2012**; **Ozeki et al., 2009**). Indeed, taking the coupling $J$ to be weak allows the

expansion $(1 + J_I L_I - J_E L_E)^{-2} \approx 1 + 2 J_E L_E - 2 J_I L_I$ in **Equation (7)**, so that the attention mediated increase in $L_I$ reduces population variance through cancellation, as in **Equation (5)**. However, this expansion is not formally required to compute the eigenvalues $\lambda_1$ and $\lambda_2$, and these measure the stability of the firing rate dynamics. We mention the expansion only to compare to the original motivation for inhibition.

The expression for $V_E$ given above (**Equation 7**) assumes a symmetry in the network coupling, namely that $J_{EE} = J_{IE} \equiv J_E$ and $J_{EI} = J_{II} \equiv J_I$. This allowed $V_E$ to be compactly written, facilitating the analysis of how attention affects both the numerator and denominator of **Equation (7)**. However, the linearization of the mean field equations and the subsequent analysis of population variability do not require this assumption (see Materials and methods: Mean field model **Equations (18–20)**). To explore the robustness of our main result we let $J_{IE} = \alpha J_E$ and $J_{II} = \beta J_I$, thereby breaking the coupling symmetry for $\alpha, \beta \neq 1$. The reduction in $V_E$ with attention is robust over a large region of $(\alpha, \beta)$ (**Figure 6a**, green region). Focusing on selected $(\alpha, \beta)$ pairings within the region where $V_E$ decreases shows that the attentional path identified for the network with coupling symmetry produces qualitatively similar behavior in the more general network (compare **Figure 5c** to **Figure 6b–e**). In total, the

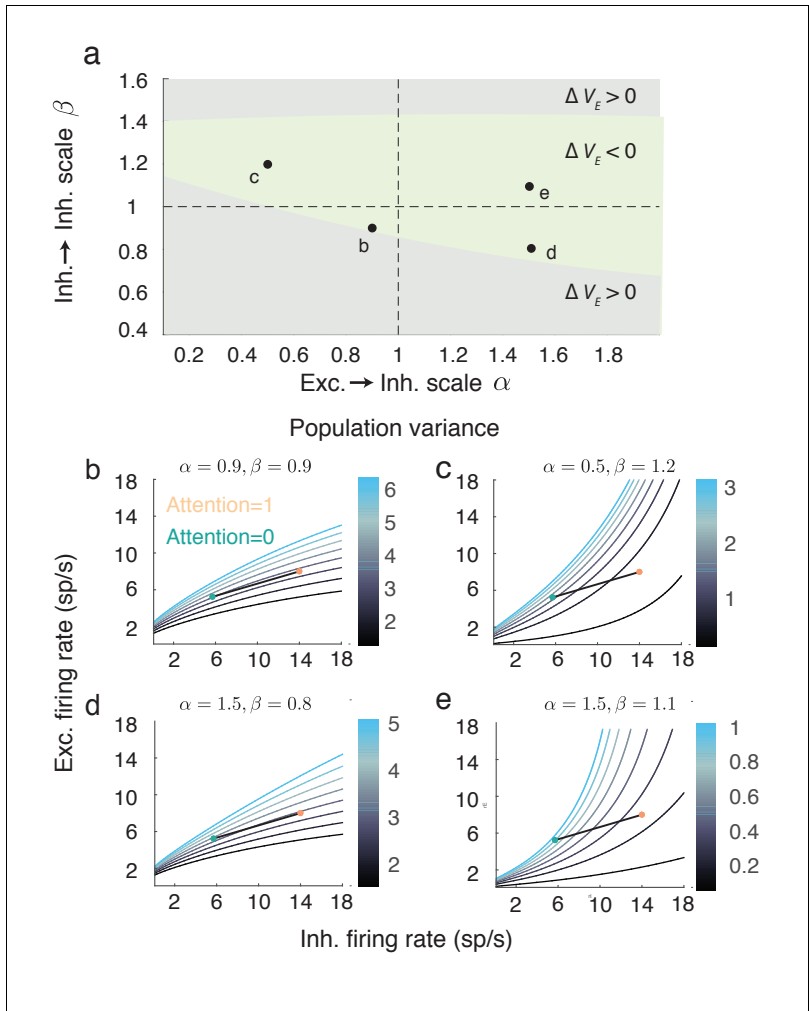

**Figure 6.** The attention mediated reduction in population variance is robust to changes in strength of recurrent connectivity. (a) Sweep over $\alpha = J_{EE}/J_{IE}$ and $\beta = J_{EI}/J_{II}$ space (with $J_{EE}$ and $J_{EI}$ fixed) labeling the region where $\Delta V_E = V_E^U - V_E^A$ is positive (grey) and negative (green). (b–e) Attentional path in excitatory-inhibitory firing rate space. The colored contours are as in **Figure 5a**. All calculations are done using **Equations (18–20)**.

inhibitory mechanism for attention mediated reduction in population variability is robust to changes in the recurrent coupling with the network.

While the reduced mean field equations are straightforward to analyze, a similar attenuation of pairwise covariance $\mathrm{Cov}(y_i, y_j)$ along the same attentional path occurs in the LIF model network (Appendix: Spiking network). Using linear response analysis for the spiking network we can relate the effect of inhibition to previous work in spiking networks (*Renart et al., 2010*; *Tetzlaff et al., 2012*; *Ly et al., 2012*; *Doiron et al., 2016*). In particular, the attention-mediated decrease of $\mathrm{Cov}(y_i, y_j)$ occurs for a wide range of timescale, ranging as low as 20 ms. However, for short time-scales that match the higher gamma frequency range (approximately 60–70 Hz) this attentional modulation increases $\mathrm{Cov}(y_i, y_j)$ (*Appendix 1—figure 6*). This finding is consistent with reports of attention-mediated increases of neuronal synchrony on gamma frequency timescales(*Fries et al., 2001*; *Buia and Tiesinga, 2008*), particularly when inhibitory circuits are engaged (*Kim et al., 2016*).

## Attention can simultaneously increase stimulus gain and decrease noise covariance

An important neural correlate of attention is enhanced stimulus response gain (*McAdams and Maunsell, 2000*). The previous section outlines how the recruitment of recurrent inhibitory feedback by attention reduces response variability. However, inhibitory feedback is also a common gain control mechanism, and increased inhibition reduces response gain through the same mechanism that dampens population variability (*Sutherland et al., 2009*). Thus it is possible that the decorrelating effect of attention in our model may also reduce stimulus response gain as well, which would make the model inconsistent with experimental data.

To insert a bottom-up stimulus $s$ in our model we let the attention-independent background input have a stimulus term: $\mu_{\alpha B} = k_\alpha s + \hat\mu_{\alpha B}$. Here $k_\alpha$ is the feedforward stimulus gain to population $\alpha$ and $\hat\mu_{\alpha B}$ is the background input that is both attention and stimulus independent. Our model captures a bulk firing rate $r_E$ rather than a population model with distributed tuning. Because of this the stimulus $s$ should either be conceived as the contrast of an input, or the population conceived as a collection of identically-tuned neurons (i.e a single cortical column).

Straightforward analysis shows that the stimulus response gain of the excitatory population can be written as (Materials and methods: Computing stimulus response gain):

$$G_E \equiv \frac{d\bar r_E}{ds} = \frac{k_E \sqrt{V_E}}{\sigma_E} + \frac{J_I L_E L_I}{1 + J_I L_I - J_E L_E}(k_E - k_I). \tag{8}$$

If $k_E = k_I$ then $G_E \propto \sqrt{V_E}$, and thus any attentional modulation that reduces population variability will necessarily reduce population stimulus sensitivity. However, for $k_E > k_I$ the second term in *Equation (8)* can counteract this effect and decouple stimulus sensitivity and variability modulations.

Consider the example attentional path (*Figure 4g*) with the extreme choice of $k_E = 1$ and $k_I = 0$. In this case attention causes an increase in $G_E$ (*Figure 7a,b*), while simultaneously causing a decrease in $V_E$ (*Figure 5a,b*). This is a robust effect, as seen by the region in $(\bar r_E, \bar r_I)$ space for which the change in $V_E$ from the unattended state is negative, and the change in $G_E$ is positive (green region, *Figure 7c*). Further, for fixed $k_I$ the proportion of the gray rectangle that the green region occupies increases with $k_E > k_I$ (*Figure 7d*). Thus, the decoupling of attentional effects on population variability and stimulus sensitivity is robust to both attentional path $(\Delta\mu_E, \Delta\mu_I)$ and feedforward gain $(k_E, k_I)$ choices. The condition that $k_E > k_I$ implies that feedforward stimuli must directly target excitatory neurons to a larger degree than inhibitory neurons (or at least the inhibitory neurons subject to attentional modulation). This gives us a complementary prediction to the one from the previous section: while top-down attention favors inhibitory neurons, the bottom-up stimulus favors excitatory neurons.

In total, our model of attentional modulation in recurrently coupled excitatory and inhibitory cortical networks subject to global fluctuations satisfies three main neural correlates of attention: (1) increase in excitatory firing rates and in (2) stimulus-response gain, with a (3) decrease in pairwise excitatory neuron co-variability.

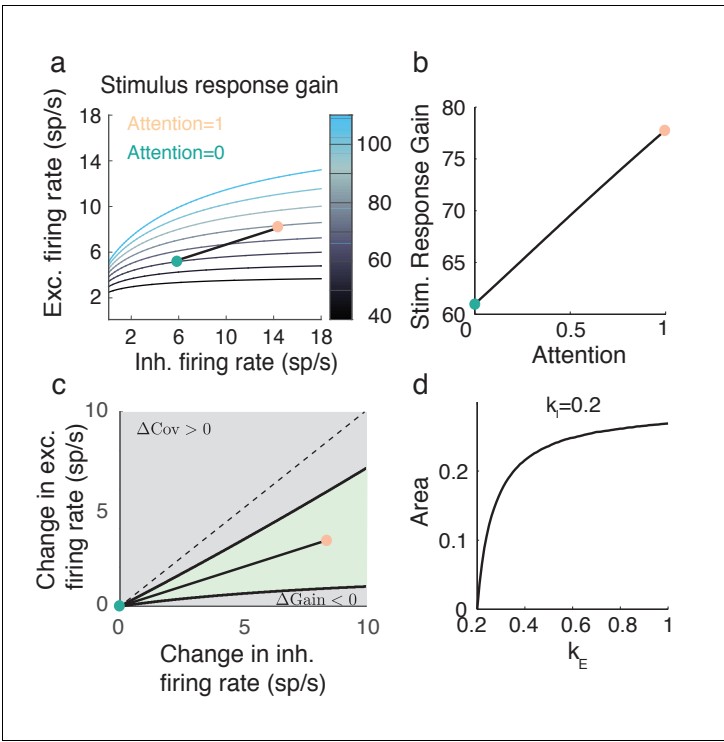

**Figure 7.** Attention model can capture increase in stimulus response gain $G_E$ despite decrease in population variance $V_E$. (a) Attentional path through $(\bar{r}_E, \bar{r}_I)$ space shows an increase in stimulus response gain. The shown path is the same path as in *Figure 5*. (b) Values of $G_E$ along the path depicted in a. (c) The green region in $(\bar{r}_E, \bar{r}_I)$ space denotes where $\Delta V_E = \mathrm{Var}^A(r_E) - \mathrm{Var}^U(r_E) < 0$ and $\Delta G_E = G_E^A - G_E^U > 0$. Black lines are iso-lines of covariance and gain, along which those quantities do not change. (d) Percent area of the green region in c out of a constant rectangle, as the feedforward stimulus gain $k_E$ increases, with $k_I = 0.2$ held constant.

## Impact of attentional modulation on neural coding

Attention serves to enhance cognitive performance, especially on discrimination tasks that are difficult (*Moore and Zirnsak, 2017*). Thus, it is expected that the attention-mediated reduction in population variability and increase in stimulus response gain subserve an enhanced stimulus estimation (*Cohen and Maunsell, 2009*; *Ruff and Cohen, 2014*). In this section we investigate how the attentional modulation outlined in the previous sections affects stimulus coding by the population.

As mentioned above our simplified mean field model (*Equation 6*) considers only a bulk response, where any individual neuron tuning is lost. As such a proper analysis of population coding is not possible. Nonetheless, our model has two basic features often associated with enhanced coding, decreased population variability (*Figure 5*) and increased stimulus-response gain (*Figure 7*).

Fisher information (*Averbeck et al., 2006*; *Beck et al., 2011*) gives a lower bound on the variance of a stimulus estimate constructed from noisy population responses, and is an often used metric for population coding. The linear Fisher information (*Beck et al., 2011*) $FI_{EI}$ computed from our two-dimensional recurrent network is:

$$\mathrm{FI}_{EI} = \begin{bmatrix} G_E & G_I \end{bmatrix} \begin{bmatrix} V_E & C_{EI} \\ C_{EI} & V_I \end{bmatrix}^{-1} \begin{bmatrix} G_E \\ G_I \end{bmatrix} = \mathrm{constant} \tag{9}$$

Here $V_\alpha = \mathrm{Var}(r_\alpha)$, $G_\alpha = d\bar{r}_\alpha/ds$, and $C_{EI} = \mathrm{Cov}(r_E, r_I)$. The important result is that $FI_{EI}$ is invariant with attention, meaning that attention does not increase the network's capacity to estimate the stimulus $s$.

While the proof of *Equation (9)* is straightforward and applies to our recurrent excitatory-inhibitory population (see Materials and methods: Fisher information), the invariance of the total information $F_{EI}$ with attention is most easily understood by analogy with an uncoupled, one-dimensional

excitatory population (*Figure 8a*). Without coupling, the input to the population is simply $k_E s + \sigma_E \xi(t)$, which is then passed through the firing rate nonlinearity $f_E$. In this case the gain is $G_E = k_E L_E$, and assuming a linear transfer the population variance is $V_E = \sigma_E^2 L_E^2$. In total the linear Fisher information from the uncoupled population is then:

$$FI_E^{uc} = \frac{G_E^2}{V_E} = \frac{(k_E L_E)^2}{\sigma_E^2 L_E^2} = \frac{k_E^2}{\sigma_E^2}. \tag{10}$$

The proportion $L_E^2$ by which attention increases the squared gain (*Figure 8a*, top) is exactly matched by the attention related increase in population variance (*Figure 8a*, bottom), resulting in cancellation of any attention-dependent terms in $FI_E$.

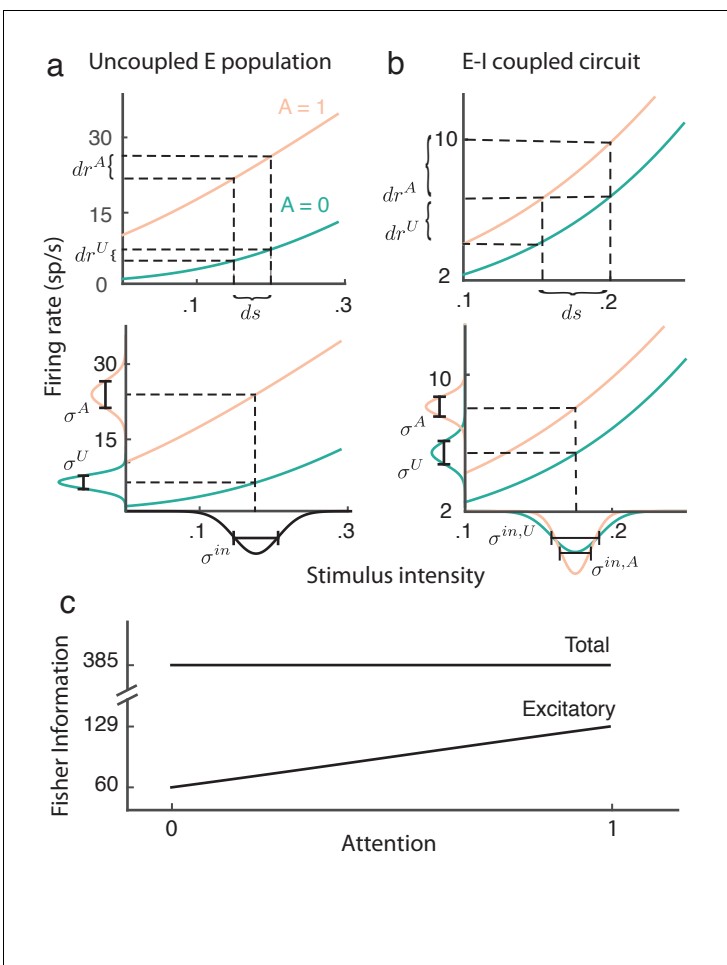

**Figure 8.** Attention improves stimulus estimation by the excitatory population embedded within excitatory (*E*)-inhibitory (*I*) network. (a) Top: For a uncoupled excitatory population, the stimulus response gain $G_E$ increases with attention. Turquoise: unattended state; orange: attended state. Bottom: Population variance $V_E$ increases with attention. Stimulus-response curves same as above. Input variance is computed from all input to a population, including external noise and recurrent coupling. The Fisher information for the uncoupled *E* population is constant with attention because the squared gain $G_E^2$ and variance $V_E$ increase proportionally (b) Same as (a) but for the *E* population within the $E - I$ network. Top: $G_E$ increases with attention. Bottom: $V_E$ decreases with attention, because the net input variance of the *E* population decreases with attention. (c) Total Fisher information for coupled E-I populations is constant with attention. By contrast, the Fisher information of the excitatory component $FI_E$ increases with attention.

The majority of projection neurons in the neocortex are excitatory, so we now consider the stimulus estimation from a readout of only the excitatory population. Combining our previous results we obtain:

$$FI_E = \frac{G_E^2}{V_E} = \frac{(J_I L_I (k_E - k_I) + k_E)^2}{\sigma_E^2 - J_I^2 L_I^2 (\sigma_E \sigma_I - \sigma_E^2 - \sigma_I^2) - 2 J_I L_I \sigma_E (\sigma_I - \sigma_E)}. \tag{11}$$

Restricting the readout to be from only the excitatory population drastically reduces the total information (compare $FI_{EI}$ to $FI_E$ in *Figure 8c*). As with the uncoupled population the response gain $G_E$ of the excitatory neurons in the coupled population increases with attention (*Figure 8b*, top). Yet unlike the uncoupled population the net input variability to the $E$ population is reduced by attention through a cancelation of the external variability $\xi(t)$ via inhibition (*Figure 8b*, bottom). These two components combine so that despite $\mathrm{FI}_E < \mathrm{FI}_{EI}$, we have that $\mathrm{FI}_E$ *does* increase with attention (*Figure 8c*). In sum, even though the total stimulus information in the network does not change with attention, the amount of information extractable from the excitatory population increases, which could lead to improved downstream stimulus estimation in the attended state.

## Discussion

Using population recordings from visual area V4 we identified rank one structure in the mapping of population spike count covariability between unattended and attended states. We used this finding to motivate an excitatory-inhibitory cortical circuit model that captures both the attention-mediated increases in the firing rate and stimulus response gain, as well as decreases in noise correlations. Our model accomplishes this with only an attention dependent shift in the overall excitability of the cortical population, in contrast to a scheme where distinct biophysical mechanisms would be responsible for respective firing rate and noise correlations modulations. The model makes two key predictions about how stimulus and modulatory inputs are distributed over the excitatory-inhibitory cortical circuit. First, top-down attentional signals must affect inhibitory neurons more than excitatory neurons to allow a better damping of global fluctuations in the attended state. Second, bottom-up stimulus information must be biased towards excitatory cells to permit higher gain in the attended state. In total, the increased response gain and decreased correlations enhance the flow of information when the readout is confined to the excitatory population.

### Candidate physiological mechanisms for attentional modulation

Our model does not consider a specific type of inhibitory neuron, and rather models a generic recurrent excitatory-inhibitory circuit. However, inhibitory circuits in cortex are complex, with at least three distinct interneuron types being prominent in many areas: parvalbumin- (PV), somatostatin- (SOM), and vasointestinal peptide-expressing (VIP) interneurons (*Rudy et al., 2011*; *Pfeffer et al., 2013*; *Kepecs and Fishell, 2014*). In mouse visual cortex, both SOM and PV cells form recurrent circuits with pyramidal cells, with PV cells having stronger inhibitory projections to pyramidal cells than those of SOM cells (*Pfeffer et al., 2013*). Furthermore, PV and SOM neurons directly inhibit one another, with the SOM to PV connection being stronger than the PV to SOM connection (*Pfeffer et al., 2013*). Finally, VIP cells project strongly to SOM cells (*Pfeffer et al., 2013*) and are activated from inputs outside of the circuit (*Lee et al., 2013*; *Fu et al., 2014*), making them an attractive target for modulation. Recent studies in visual, auditory, and somatosensory cortical circuits show that VIP cell activation provides an active disinhibition of pyramidal cells via a suppression of SOM cells (*Kepecs and Fishell, 2014*). Basal forebrain (BF) stimulation modulates both muscarinic and nicotinic ACh receptors (mAChRs and nAChRs respectively) in a fashion that mimics attentional modulation (*Alitto and Dan, 2012*). In particular, the recruitment of VIP cell activity in vivo through BF stimulation is strongly dependent on both the muscarinic and nicotinic cholinergic pathways (*Alitto and Dan, 2012*; *Kuchibhotla et al., 2017*; *Fu et al., 2014*), and it has thus been hypothesized VIP cells activation could be an important component of attentional modulation (*Alitto and Dan, 2012*; *Poorthuis et al., 2014*).

If we consider the inhibitory population in our model to be PV interneurons then the recruitment of VIP cell activity via top-down cholinergic pathways is consistent with our attentional model in two ways. First, activation of the VIP → SOM → pyramidal cell pathway provides a disinhibition to

pyramidal cells, modeled simply as an overall depolarization to pyramidal cells in the attended state (*Figure 4*). Second, the activation of the VIP → SOM → PV cell pathway disinhibits PV cells, and the strong SOM → PV projection would suggest that the disinhibition is sizable as required by our model (*Figure 5c*). Finally, a recent study in mouse medial prefrontal cortex reports that identified PV interneurons show an attention related increase in activity, and that optogenetic silencing of PV neurons impairs attentional processing (*Kim et al., 2016*).

However, our logic is perhaps overly simplistic and neglects the direct modulation of SOM cells via muscarinic and nicotinic cholinergic pathways (*Alitto and Dan, 2012*; *Kuchibhotla et al., 2017*) that could compromise the disinhibitory pathways. Further, there is evidence of a direct ACh modulation of PV cells (*Disney et al., 2014*) as opposed to through a disinhibitory pathway. Finally, there may be important differences across both species (mouse vs. primate) and visual area (V1 vs. V4) that fundamentally change the pyramidal, PV, SOM, and VIP circuit that is understood from mouse V1 (*Pfeffer et al., 2013*). Future studies in the inhibitory to excitatory circuitry of primate visual cortex, and its attentional modulation via neuromodulation, are required to navigate these issues.

Finally, the simultaneous increase in response gain and decrease in noise correlations with attention requires excitatory neurons to be more sensitive to bottom-up visual stimulus than inhibitory neurons ($k_E > k_I$, *Figure 7*). In mouse visual cortex, GABAergic interneurons show overall less stimulus selectivity than pyramidal neurons (*Sohya et al., 2007*), however this involves both direct feedforward and recurrent contributions to stimulus tuning. While our model simplified the feedforward stimulus gain $k_E$ and $k_I$ to be constant with attention, it is known that attention also modulates feedforward gain through presynaptic nACh receptors (*Disney et al., 2007*). Notably, nAChRs are found at thalamocortical synapses onto layer 4 excitatory cells and not onto inhibitory neurons, suggesting that $k_E$ would increase with attention while $k_I$ would not. Thus, $k_E$ should also increase with attention while $k_I$ should not, further supporting that $k_E > k_I$.

## Modeling global network fluctuations and their modulation

Our model considered the source of global fluctuations as external to the network. This choice was due in part to difficulties in producing global, long timescale fluctuations through strictly internal coupling (*Renart et al., 2010*; *Rosenbaum et al., 2017*). Our model assumed that the intensity of these external input fluctuation were independent of attention. Rather, attention shifted the operating point of the network such that the transfer of input variability to population-wide output activity was attenuated in the attended state.

Recent analysis of population recordings show that generative models of spike trains that consider gain fluctuations in conjunction with standard spike emission variability capture much of the variability of cortical dynamics (*Rabinowitz et al., 2015*; *Lin et al., 2015*). Further, these gain fluctuations are well approximated by a one-dimensional, global stochastic process affecting all neurons in the population (*Ecker et al., 2014*; *Rabinowitz et al., 2015*; *Lin et al., 2015*; *Ecker et al., 2016*; *Engel et al., 2016*; *Whiteway and Butts, 2017*). When these techniques are applied to population recordings subject to attentional modulation, the global gain fluctuations are considerably reduced in the attended state (*Rabinowitz et al., 2015*; *Ecker et al., 2016*). Our assumption that external input fluctuations to our network are attention-invariant is consistent with this statistical analysis since it is necessarily constructed from only output activity. Nevertheless, another potential model is that the reduction in population variability is simply inherited from an attention-mediated suppression of the global input fluctuations. Unfortunately, it is difficult to distinguish between these two mechanisms when restricted to only output spiking activity.

However, a model where output variability reductions are simply inherited from external inputs suffers from two criticisms. First, it begs the question: what is the mechanism behind the shift in input variability? Second, our model requires only an increase in the external depolarization to excitatory and inhibitory populations to account for all attentional correlates. An inheritance model would necessarily decouple the attentional mechanisms behind increases in network firing rate (still requiring a depolarization) and the decrease in global input variability. Thus, our model offers a parsimonious and biologically motivated explanation of these neural correlates of attention. Further work dissecting the various external and internal sources of variability to cortical networks, and their attentional modulation, is needed to properly validate or refute these different models.

## Attentional modulation of neural coding through inhibition

Our network model assumed attention-invariant external fluctuations and weak recurrent inputs, permitting a linear analysis of network activity. As a consequence the linear information transfer by the entire population was attention-invariant (*Figure 8*), because attention modulated the network's transfer of signal and noise equivalently. However, this invariance was only apparent if the decoder had access to both the excitatory and inhibitory populations. However, most of the neurons in cortex that project between areas are excitatory. When the decoder was restricted to only the activity of the excitatory population then our analysis uncovered two main results. First, the excitatory population carried less information than the combined excitatory-inhibitory activity, suggesting an inherently suboptimal coding scheme used by the cortex. Second, the attention-mediated modulation of the inhibitory neurons increased the information carried by the excitatory population. This agrees with the wealth of studies that show that attention improves behavioral performance on stimulus discrimination tasks.

Determining the impact of population-wide spiking variability on neural coding is complicated (*Averbeck et al., 2006*; *Kohn et al., 2016*). A recent theoretical study has shown that noise correlations that limit stimulus information must be parallel to the direction in which population activity encodes the stimulus (*Moreno-Bote et al., 2014*). The fluctuations in our network satisfy this criteria, albeit trivially since all neurons share the same stimulus input. Indeed, in our network the external inputs appear to the network as $s + x(t)$, meaning that fluctuations from the noise source $x(t)$ are indistinguishable from fluctuations in the stimulus $s$. This is an oversimplified view and assumes that the decoder treats the neurons as indistinguishable from one another, at odds with classic work in population coding (*Pouget et al., 2000*). Extending our network to include distributed tuning and feature-based recurrent connectivity is a natural next step (*Ben-Yishai et al., 1995*; *Rubin et al., 2015*). To do this the spatial scales of feedforward tuning, recurrent projections, external fluctuations, as well as attention modulation must all be specified. It is not clear how noise correlations will depend on these choices yet work in spatially distributed balanced networks shows that solutions can be complex (*Rosenbaum et al., 2017*).

The role of inhibition in shaping cortical function is a longstanding topic of study (*Isaacson and Scanziani, 2011*), including recent work showing inhibition can actively decorrelate cortical responses (*Renart et al., 2010*; *Tetzlaff et al., 2012*; *Ly et al., 2012*). Our work gives a concrete example of how this decorrelation can be gated and used to control the flow of information. Of interest are tasks that probe a distributed population where attention again decreases noise correlations between neurons with similar stimulus preference, yet *increases* noise correlations between cells with dissimilar stimulus preference (*Ruff and Cohen, 2014*). The circuit mechanisms underlying this neural correlate of attention are unclear. However, there is ample work in understanding how recurrent inhibition shapes cortical activity in distributed populations (*Isaacson and Scanziani, 2011*), including in models of attentional circuits (*Ardid et al., 2007*; *Buia and Tiesinga, 2008*). Adapting our model to include distributed tuning is an important next step and will be a better framework to discuss the coding consequences of the attentional modulation circuits proposed in our study.

## Methods and materials

### Data preparation

Data was collected by from two rhesus monkeys with microelectrode arrays implanted bilaterally in V4 as they performed an orientation-change detection task (*Figure 1a*) (*Cohen and Maunsell, 2009*). All animal procedures were in accordance with the Institutional Animal Care and Use Committee of Harvard Medical School. Two oriented Gabor stimuli flashed on and off several times, until one of them changed orientation. The task of the monkey was to then saccade to the stimulus that changed. Each recording session consisted of at least four blocks of trials in which the monkey's attention was cued to the left or right. We excluded from the analysis instruction trials which occurred at the start of each block to cue the monkey to one side to attend to, catch trials in which the monkey was rewarded just for fixating, and trials in which the monkey did not perform the task correctly. Moreover, the first and last stimulus presentations in each trial were not analyzed, to prevent transients due to stimulus appearance or change from affecting the results. The total number of

trials included in the analysis from all the recording sessions was $42,496$. Each trial consisted of between $3$ and $12$ stimulus presentations, of which all but the first and last were analyzed.

Recordings from the left and right hemispheres of each monkey were analyzed separately because the activities of the neurons in opposite hemispheres had near-zero correlations (*Cohen and Maunsell, 2009*). Neurons in the right hemisphere were considered to be in the attended state when the attentional cue was on the left, and vice-versa. We note that because our criteria for choosing which trials and units to analyze were based on different needs for data analysis compared to the original study (*Cohen and Maunsell, 2009*) the specific firing rates and covariances differ quantitatively from those previously reported.

In monkey 1, an average of 51.1 (min 35, max 80) units were analyzed from the right hemisphere, and an average of 27.5 (min 14, max 56) units were analyzed from the left hemisphere. From monkey 2, an average of 56.6 (min 43, max 71) units from the right hemisphere, and an average of 37.7 (min 32, max 46) units from the left hemisphere were analyzed. From each recording, spikes falling between 60 and 260 ms from stimulus onset were considered for the firing rate analysis, to account for the latency of neuronal responses in V4.

## Comparing change in covariance to change in variance

Let $S^U$ be the matrix containing spike counts of the neurons on trials in which they are in the unattended state, and $S^A$ the matrix containing spike counts of the neurons on trials in which they are in the attended state. Denote the unattended spike count covariance matrix by $C^U = \mathrm{Cov}(S^U)$, and the attended one by $C^A = \mathrm{Cov}(S^A)$. Attentional changes in covariance and variance were measured both on average (*Figure 1c*) and as distributions (*Figure 1d*). The distributions of the normalized differences

$$\frac{\mathrm{Cov}^A - \mathrm{Cov}^U}{\max(|\mathrm{Cov}^A|,|\mathrm{Cov}^U|)} \quad \text{and} \quad \frac{\mathrm{Var}^A - \mathrm{Var}^U}{\max(|\mathrm{Var}^A|,|\mathrm{Var}^U|)} \tag{12}$$

reveal a concentration of negative covariance changes, and a distribution of variance changes symmetric about zero. Here, $\mathrm{Cov}^A$ and $\mathrm{Cov}^U$ ($\mathrm{Var}^A$ and $\mathrm{Var}^U$) are vectors containing covariance (variance) values of the entire data set. Note that the distributions are bounded between $-2$ and $2$ by construction.

## Solving systems of equations by error minimization

When solving systems of the form of *Equation (2)* in order to quantify the fit of the model, a nonlinear equation solver (fminunc) in MATLAB was used. The solver found minima of an objective function which we defined as the Euclidean norm of the difference of the approximation of the attended covariance matrix and the original attended covariance matrix, in other words, the error of the approximation:

$$f(g_1,...,g_N) = \sqrt{\sum_{i<j}(g_i C^U(i,j)g_j - C^A(i,j))^2}. \tag{13}$$

## Shuffled covariance matrices

For finite population sizes ($N < \infty$) we expect our algorithm to extract some low-rank structure between arbitrary covariance matrices. Let $\sqrt{C^A}$ be the principal square root of the attended covariance matrix, the unique positive-semidefinite square root of a positive-semidefinite matrix. Consider the symmetric matrix $D = \mathrm{perm}(\sqrt{C^A})$ computed from the a random permutation of the upper-triangular entries of $\sqrt{C^A}$. Finally, let $C^A_{\mathrm{shuf}} = \mathrm{real}(DD)$. The square root-permutation-squaring procedure guarantees a positive-semidefinite matrix, as the square of any matrix is positive-semidefinite. Shuffling removes any relation between $\mathbf{C^U}$ and $C^A_{shuf}$, and any remaining detected structure would be due to finite sampling. The shuffled covariance gain $\hat{g}_{\mathrm{shuf}}$ provides the prediction $\hat{C}^A_{\mathrm{shuf}} := \hat{g}_{\mathrm{shuf}}\hat{g}^T_{\mathrm{shuf}} \circ C^U$, and $\rho_{\mathrm{shuf}}$ measures the relation between $\hat{C}^A_{\mathrm{shuf}}$ and $C^A_{\mathrm{shuf}}$. Synthetic data shows that as population size $N$ becomes large the coefficient $\rho_{\mathrm{shuf}}$ approaches 0 (Appendix: Detected structure in random covariance matrices is a finite-size effect).

## Upper bound covariance matrices

The covariance matrices $\mathbf{C^U}$ and $\mathbf{C^A}$ are estimates obtained from a finite number of trials, and any estimation error will compromise the ability to detect rank one structure of $A_C$. Here we outline an upper bound for the model performance based on a finite number of trials over which the covariance matrices were originally estimated. Let $\hat{C}^A := \hat{\mathbf{g}}\hat{\mathbf{g}}^T \circ C^U$ with $\hat{\mathbf{g}}$ minimizing the $L^2$ norm of $C^A := \mathbf{g}\mathbf{g}^T \circ C^U$. We remark that $\hat{C}^A$ perfectly decomposes according to the statistical model in *Equation (2)*. We used $\hat{C}^A$ to generate an artificial set of $N$ correlated Poisson spike counts, using an algorithm based on a latent multivariate gaussian model (*Macke et al., 2009*). We sampled these population spike counts with a fixed number of trials ($M$) with $D$ be the resulting $M \times N$ matrix of Poisson samples for each process. Let $C^A_{\mathrm{ub}} = \mathrm{Cov}(D)$ be the 'upper bound' covariance matrix: a finite trial sampling approximation to the perfectly decomposable matrix $\hat{C}^A$. Finally, we employ our algorithm to give $\hat{C}^A_{\mathrm{ub}} := \hat{\mathbf{g}}_{\mathrm{ub}}\hat{\mathbf{g}}^T_{\mathrm{ub}}C^U$, where the vector $\hat{\mathbf{g}}_{\mathrm{ub}}$ minimizes the $L^2$ norm of the error.

Since $\hat{C}^A$ is perfectly decomposable then for $M \to \infty$ we have $\hat{C}^A_{\mathrm{ub}} = C^A_{\mathrm{ub}} = \hat{C}^A$. Thus in the large $M$ limit the coefficient $\rho_{\mathrm{ub}}$ between elements of $\hat{C}^A_{\mathrm{ub}}$ and $C^A_{\mathrm{ub}}$ converges to 1 (Appendix: Performance limited by available number of trials). However, for finite $M$ we have that $\rho_{\mathrm{ub}} < 1$, solely due to inaccuracies in estimating $\hat{C}^A$ with $C^A_{\mathrm{ub}}$. To account for the possibility of particular strings of realizations $D$ introducing random biases into $C^A_{\mathrm{ub}}$, we performed the following analysis on 10 independently generated upper-bound covariance matrices $C^A_{\mathrm{ub}}$.

## Leave-one-out cross-validation

Instead of solving the system consisting of all *Equations (2)*, we remove one of them. Denote the complete set of equations by $S$, an individual equation as $s_{ij} := \{C^A_{ij} = g_i g_j C^U_{ij}\}$ and the set of equations with one of them removed as $S_{ab} := S - s_{ab}$. We then solve the system $S_{ab}$. Denote the solution by $\mathbf{g}_{ab}$. We can then compare $C^A_{ab}$ and $\hat{C}^A_{ab} = \mathbf{g}_{ab}(a)\mathbf{g}_{ab}(b)C^U_{ab}$. We do this for $\max(1000, N(N-1)/2)$ possible systems $S_{ab}$. The $\rho$ of the vector of resulting $C^A_{ab}$ vs $\hat{C}^A_{ab}$ values is a measure of how well the system can predict one of its elements, or in other words, how well the structure holds together when one element is taken out. This leave-one-out cross-validation was performed for the shuffled and the upper-bound cases as well.

## Mean field model

The mean spiking activity over the population $\alpha$ $(= E \text{ or } I)$ is

$$r_\alpha(t) = \langle y_{i\alpha}(t)\rangle_i, \tag{14}$$

where $y_{i\alpha}(t) = \sum_{j=1}^{n_{i\alpha}} \delta(t - t^j_{i\alpha})$ is the spike train of excitatory neuron $i$ of population $\alpha$, $n_{i\alpha}$ is the number of spikes from that neuron, and $t^j_{i\alpha}$ is the time of spike $j$. We follow previous studies (*Tetzlaff et al., 2012*; *Ozeki et al., 2009*; *Ledoux and Brunel, 2011*) and consider the firing rate dynamics of the $E$ and $I$ populations given by the system in *Equations (6)*:

$$\tau_E \frac{dr_E}{dt} = -r_E + f_E\left(\mu_{EB} + A\Delta\mu_E + J_{EE}r_E - J_{EI}r_I + \sigma_E\left[\sqrt{1-\chi}x_E(t) + \sqrt{\chi}x(t)\right]\right),$$

$$\tau_I \frac{dr_I}{dt} = -r_I + f_I\left(\mu_{IB} + A\Delta\mu_I + J_{IE}r_E - J_{II}r_I + \sigma_I\left[\sqrt{1-\chi}x_I(t) + \sqrt{\chi}x(t)\right]\right).$$

Here $\mu_{\alpha B}$ is the attention independent drive to population $\alpha$, $A \in [0, 1]$ is the attention variable, and $\Delta\mu_\alpha$ is the maximal drive to population $\alpha$ due to attention. The parameter $J_{\alpha\beta}$ is the coupling from population $\beta$ to populations $\alpha$. The stochastic processes $x_E(t)$, $x_I(t)$, and $x(t)$ are the global fluctuations applied to the network. The excitatory and inhibitory populations have private fluctuations $x_\alpha(t)$ and also common fluctuations $x(t)$ given to both populations; the parameter $\chi$ scales the degree of private versus common fluctuations. We perform calculations for arbitrary $\chi$ and then take $\chi \to 1$ to match the system given in *Equations (6)*. The total intensity of fluctuations to population $\alpha$ is set by $\sigma_\alpha$. These simplified rate equations give an accurate picture of the long-timescale dynamics of networks of coupled spiking neuron models that are in the fluctuation driven regime (*Ledoux and Brunel, 2011*). The operative timescale reflects a combination of synaptic and membrane

integration; since we are interested in spiking covariance over time windows that are much longer than these, we take them to be unity for simplicity.

To give a quantitative match between the equilibrium statistics of the rate equations and the leaky integrate-and-fire (LIF) network simulations we take the transfer function $f$ to be the inverse first passage time of an LIF neuron driven by white noise (*Ledoux and Brunel, 2011*):

$$f_\alpha(I) = \left( \tau_\alpha \sqrt{\pi} \int_{(-V_T+I)/\eta_\alpha}^{(-V_R+I)/\eta_\alpha} \exp(z^2)\mathrm{erfc}(z)dz \right)^{-1}. \tag{15}$$

The parameter $\eta_\alpha$ is the intensity of the external fluctuations given to the LIF neurons (Appendix: Spiking model). The membrane timescale $\tau$ gives the dimensions of 1/s to the firing rate $r_\alpha$. The parameter $V_T$ denotes spike threshold while $V_R$ is the reset potential. Model parameters are given in *Table 1*.

If the input fluctuations, $x(t)$, $x_E(t)$, and $x_I(t)$ are white noise processes then the nonlinearity in $f$ makes the stochastic dynamics of $r_E(t)$ and $r_I(t)$ complicated (non-diffusive). To simply the analysis we consider $x(t)$ as the limiting process from:

$$\tau_x \frac{dx}{dt} = -x + \sqrt{\tau_x}\xi_x(t),$$

for $\tau_x \to 0$, with $\langle \xi_x(t) \rangle = 0$ and $\langle \xi_x(t)\xi_x(t') \rangle = \delta(t - t')$. This makes $x(t)$ sufficiently smooth in time (the same is true for $x_E(t)$ and $x_I(t)$).

We restrict the coupling $J_{\alpha\beta}$ such that for $\sigma_\alpha = 0$ the equilibrium point $(\bar{r}_E, \bar{r}_I)$ is stable and given by:

$$\bar{r}_E = f_E(\mu_{EB} + A\Delta\mu_E + J_{EE}\bar{r}_E - J_{EI}\bar{r}_I),$$
$$\bar{r}_I = f_I(\mu_{IB} + A\Delta\mu_I + J_{IE}\bar{r}_E - J_{II}\bar{r}_I). \tag{16}$$

For sufficiently small $\sigma_\alpha$ the fluctuations in population activity about the equilibrium firing rate, $\delta r_\alpha(t) = r_\alpha(t) - \bar{r}_\alpha$, obey the linearized stochastic system:

$$\tau_E \frac{d}{dt}\delta r_E = (-1 + L_E J_{EE})\delta r_E - L_E J_{EI}\delta r_I + L_E\sigma_E(\sqrt{1-\chi}x_E(t) + \sqrt{\chi}x(t)),$$
$$\tau_I \frac{d}{dt}\delta r_I = L_I J_{IE}\delta r_E - (1 + L_I J_{II})\delta r_I + L_I\sigma_I(\sqrt{1-\chi}x_I(t) + \sqrt{\chi}x(t)). \tag{17}$$

**Table 1.** Model Parameters.

| Parameter | Description | Value |
|---|---|---|
| $\tau$ | Time constants for membrane dynamics | 0.01 s |
| $V_T$ | Spike Threshold | 1 |
| $V_R$ | Spike Reset | 0 |
| $\mu_E$ | Excitatory baseline bias | 0.6089 |
| $\mu_I$ | Inhibitory baseline bias | 0.5388 |
| $\Delta\mu_E$ | Attentional modulation of excitatory bias | 0.2624 |
| $\Delta\mu_I$ | Attentional modulation of inhibitory bias | 0.3608 |
| $J_E$ | Excitatory coupling constant | 1.5 |
| $J_I$ | Inhibitory coupling constant | 3 |
| $\sigma_E$ | Amplitude of external noise to E population | 0.3 |
| $\sigma_I$ | Amplitude of external noise to I population | 0.35 |
| $c$ | Proportion of common noise to E and I populations | 1 |
| $k_E$ | Sensitivity of E population to stimulus input | 1 |
| $k_I$ | Sensitivity of I population to stimulus input | 0 |

Here $L_\alpha = \frac{df_\alpha}{dI}|_{I=I_\alpha^{\mathrm{eff}}}$ is the slope of the transfer function $f_\alpha$ evaluated at the equilibrium point $I_\alpha^{\mathrm{eff}} = \mu_\alpha + A\Delta\mu_\alpha + J_{\alpha E}\bar{r}_E - J_{\alpha I}\bar{r}_I$. **Equation (17)** is a two dimensional Ornstein-Uhlenbeck process (**Gardiner, 2004**) that is readily amenable to analysis.

## Computing $V_E$

In matrix form the system **Equation(17)** is written as:

$$\frac{d}{dt}\delta\mathbf{r} = M\delta\mathbf{r} + D\mathbf{x}. \tag{18}$$

Here $\delta\mathbf{r} = [\delta r_E, \delta r_I]$, $\mathbf{x} = [x_E, x_I, x]$, and

$$M = \begin{bmatrix} -1 + L_E J_{EE} & -L_E J_{EI} \\ L_I J_{IE} & -1 - L_I J_{II} \end{bmatrix} \text{ and } D = \begin{bmatrix} L_E\sigma_E\sqrt{1-\chi} & 0 & L_E\sigma_E\sqrt{\chi} \\ 0 & L_I\sigma_I\sqrt{1-\chi} & L_I\sigma_I\sqrt{\chi} \end{bmatrix}.$$

The stationary autocovariance function is computed as:

$$\tilde{C}(s) = \langle \delta\mathbf{r}(t), \delta\mathbf{r}(t+s) \rangle = \begin{cases} \exp(Ms)\Sigma & \text{if } s > 0 \\ \Sigma\exp(-M^T s) & \text{if } s \leq 0 \end{cases}, \tag{19}$$

where $s$ is a time lag and $\Sigma = \frac{(\mathrm{Det}M)DD^T + [M-(\mathrm{Tr}M)\mathbf{1}]DD^T[M-(\mathrm{Tr}M)\mathbf{1}]^T}{2(\mathrm{Tr}M)(\mathrm{Det}M)}$ is the variance matrix (Det and Tr denote the determinant and trace operations, respectively). Here, $\mathbf{1}$ is the $2 \times 2$ identity matrix.

The covariance between populations $\alpha$ and $\beta$ over long time scales is given by

$$C(\alpha, \beta) = \int_{-\infty}^{\infty} \tilde{C}(s; \alpha, \beta)ds, \tag{20}$$

where the integration is performed over the appropriate element of the matrix $\tilde{C}(s)$. In particular, the long timescale variance of the excitatory population is given by (after some algebra):

$$V_E = C(E, E) = \frac{L_E^2}{(1 + J_I L_I - J_E L_E)^2}(J_I L_I(\sigma_E - \sigma_I) + \sigma_E)^2. \tag{21}$$

We remark that the long timescale covariance matrix can alternatively be computed from $C = M^{-1}D[M^{-1}D]^T$ (**Gardiner, 2004**). To obtain the compact expression for $V_E$ we have assumed symmetric coupling: $J_I := J_{EI} = J_{II}$, $J_E := J_{EE} = J_{IE}$, and $\chi \to 1$. These are not required for the main results of our study and merely ease the analysis of equations.

## Computing stimulus response gain

We decompose $\mu_{\alpha B} = k_\alpha s + \hat{\mu}_{\alpha B}$ and define the gain of population $\alpha$ to stimulus $s$ as $G_\alpha = \frac{d\bar{r}_\alpha}{ds} = L_\alpha \frac{dI_\alpha}{ds}$. The term $\frac{dI_\alpha}{ds}$ is obtained by differentiating **Equations (16)**) with respect to $s$:

$$\frac{dI_\alpha}{ds} = k_\alpha + J_E G_E - J_I G_I.$$

Solving the system of two equations for $G_E$ yields:

$$G_E = \frac{L_E(k_E + J_I L_I(k_E - k_I))}{1 + J_I L_I - J_E L_E}. \tag{22}$$

For the sake of compactness we set $\sigma_E = \sigma_I$ to obtain the result in **Equation (8)**.

## Fisher information

Linear Fisher Information depends on the stimulus response gains and covariance matrix of the excitatory and inhibitory populations:

$$\text{FI}_{EI} = \begin{bmatrix} G_E & G_I \end{bmatrix} \begin{bmatrix} V_E & C_{EI} \\ C_{EI} & V_I \end{bmatrix}^{-1} \begin{bmatrix} G_E \\ G_I \end{bmatrix}$$

$$= \frac{G_E^2 V_I + G_I^2 V_E - 2 G_E G_I C_{EI}}{V_E V_I - C_{EI}^2}, \tag{23}$$

When the input correlation $0 \leq \chi < 1$ we have:

$$V_E = \left(\frac{L_E}{1 + J_I L_I - J_E L_E}\right)^2 \left(J_I^2 L_I^2(\sigma_E^2 + \sigma_I^2 - 2\sigma_E\sigma_I\chi) + 2 J_I L_I \sigma_E(\sigma_E - \sigma_I\chi) + \sigma_E^2\right), \tag{24}$$

$$V_I = \left(\frac{L_I}{1 + J_I L_I - J_E L_E}\right)^2 \left(J_E^2 L_E^2(\sigma_E^2 + \sigma_I^2 - 2\sigma_E\sigma_I\chi) + 2 J_E L_E \sigma_I(\sigma_I - \sigma_E\chi) + \sigma_I^2\right), \tag{25}$$

and

$$C_{EI} = \frac{L_E L_I}{(1 + J_I L_I - J_E L_E)^2} \quad (J_E J_I L_E L_I(\sigma_E^2 + \sigma_I^2 - 2\sigma_E\sigma_I c)$$

$$+ J_E L_E \sigma_E(\sigma_E - \sigma_I\chi) - J_I L_I \sigma_I(\sigma_I - \sigma_E\chi) + \sigma_E\sigma_I\chi). \tag{26}$$

Inserting these expressions and those for $G_E$ and $G_I$ into *Equation (23)* and simplifying yields:

$$FI_{EI} = \frac{2\chi k_E k_I \sigma_E \sigma_I - k_E^2 \sigma_I^2 - k_I^2 \sigma_E^2}{(\chi^2 - 1)\sigma_I^2 \sigma_E^2}. \tag{27}$$

We remark that $FI_{EI}$ is independent of $L_E$ and $L_I$ and thus independent of attentional modulation.

Notice that we have re-introduced the correlation constant $\chi$ into the equations, rather than only considering the limit $\chi \to 1$. If $\chi = 1$, the excitatory and inhibitory populations are receiving completely identical noise. If this is the case, the correlation cancellation would be perfect, leading to infinite informational content, as can be seen in *Equation (27)*.

## Acknowledgements

The research was supported by National Science Foundation grants NSF-DMS-1313225 (BD), NSF DMS-1517082 (BD), a grant from the Simons Foundation collaboration on the global brain (SCGB #325293MC;BD), NIH grants 4R00EY020844-03 and R01 EY022930 (MRC), a Whitehall Foundation Grant (MRC), Klingenstein-Simons Fellowship (MRC), a Sloan Research Fellowship (MRC), and a McKnight Scholar Award (MRC). We thank John Maunsell for the generous use of the data, and Kenneth Miller, Ashok Litwin-Kumar, Douglas Ruff, and Robert Rosenbaum for useful discussions.

## Additional information

### Funding

| Funder | Grant reference number | Author |
|---|---|---|
| National Science Foundation | DMS-1313225 | Tatjana Kanashiro<br>Gabriel Koch Ocker<br>Brent Doiron |
| National Science Foundation | DMS-1517082 | Gabriel Koch Ocker<br>Brent Doiron |
| Simons Foundation | Simons Collaboration on the Global Brain | Marlene R Cohen<br>Brent Doiron |
| National Institutes of Health | R01 EY022930 | Marlene R Cohen |
| National Institutes of Health | CRCNS-R01DC015139 | Brent Doiron |

The funders had no role in study design, data collection and interpretation, or the decision to submit the work for publication.

## Author contributions
TK, Formal analysis, Investigation, Writing—original draft; GKO, Formal analysis, Investigation; MRC, Conceptualization, Funding acquisition, Investigation, Writing—review and editing; BD, Conceptualization, Formal analysis, Supervision, Funding acquisition, Investigation, Writing—original draft, Writing—review and editing

## Author ORCIDs
Marlene R Cohen, http://orcid.org/0000-0001-8583-4300
Brent Doiron, http://orcid.org/0000-0002-6916-5511

## Ethics
Animal experimentation: All animal procedures were in accordance with the Institutional Animal Care and Use Committee of Harvard Medical School (Harvard IACUC protocol number: 04214).

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

**Appendix 1**

## Detected structure in random covariance matrices is a finite-size effect

Here we show that any prediction of rank one structure in our shuffled covariance matrix (non-zero $\rho_{\text{shuf}}$ in *Figure 2* of the main text) is a finite-data effect. The trial-by-trial covariance matrices of the experimental data are computed from the spike counts recorded from a set number of units. To explore the effect of population size on the detected structure in the shuffled covariance matrices we must rely on synthetic data.

We construct the synthetic covariance matrices by generating Gaussian random numbers with the same mean and standard deviation as the actual covariance matrices from the data. This construction serves as a substitute for the shuffled covariance matrices, and allows for arbitrarily large populations. As we increase the number of units from near 10 to 500, $\rho_{\text{shuf}}$ decreases accordingly, indicating that any positive $\rho_{\text{shuf}}$ is due to the finite population size, rather than any inherent structure in the data (*Appendix 1—figure 1*).

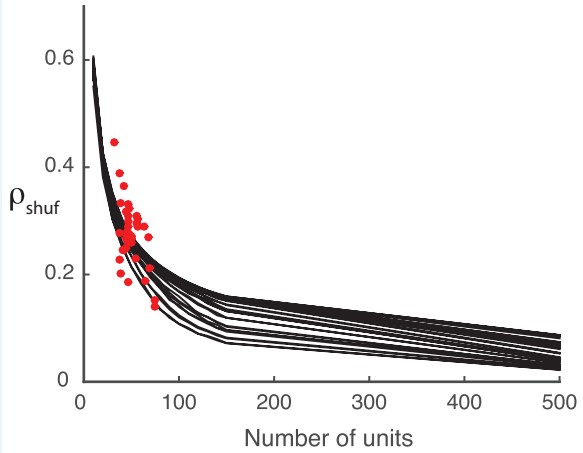

**Appendix 1—figure 1.** Detected structure in randomly generated covariance matrices is a finite-size effect. The model performance ($\rho_{\text{shuf}}$) decreases with increasing system size (black curves). The $\rho_{\text{shuf}}$ computed from the shuffled neural data (red dots) falls in the same area as the synthetic data performance, suggesting that the synthetic data is a reasonable stand-in.

## Model performance is limited by number of trials in data

The upper bound for our model $\rho_{\text{ub}}$ did not saturate 1 (see *Figure 2* of the main text). Here, we show that this is also due the finite data available. If infinitely many trials were available to compute the spike count covariance matrices from the data, and the data obeyed by the low-rank statistical model, the performance of the model ($\rho_{\text{ub}}$) should tend to one. To test this, we generate synthetic data from correlated Poisson processes as in the upper bound computation of the main text but do not limit the number of samples to the number of trials in the original data. As the number of samples increases we find that $\rho_{\text{ub}} \to 1$ (*Appendix 1—figure 2*).

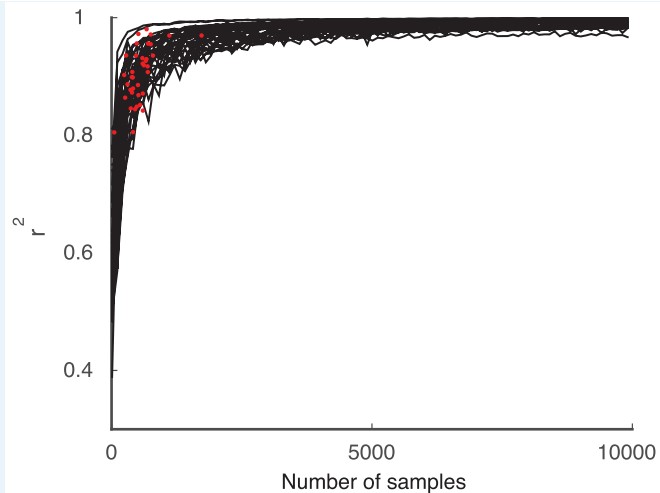

**Appendix 1—figure 2.** The performance of the model $\rho_{ub}$ (black curves) on synthetic data using increasing numbers of Poisson realizations approaches 1. The Poisson model computed with the same number of trials as the data is shown for comparison (red dots).

## Model performance for all monkeys and hemispheres

The model performance for individual recording sessions are given here for transparency (*Appendix 1—figure 3* for the full data and *Appendix 1—figure 4* for the leave-one-out cross validation).

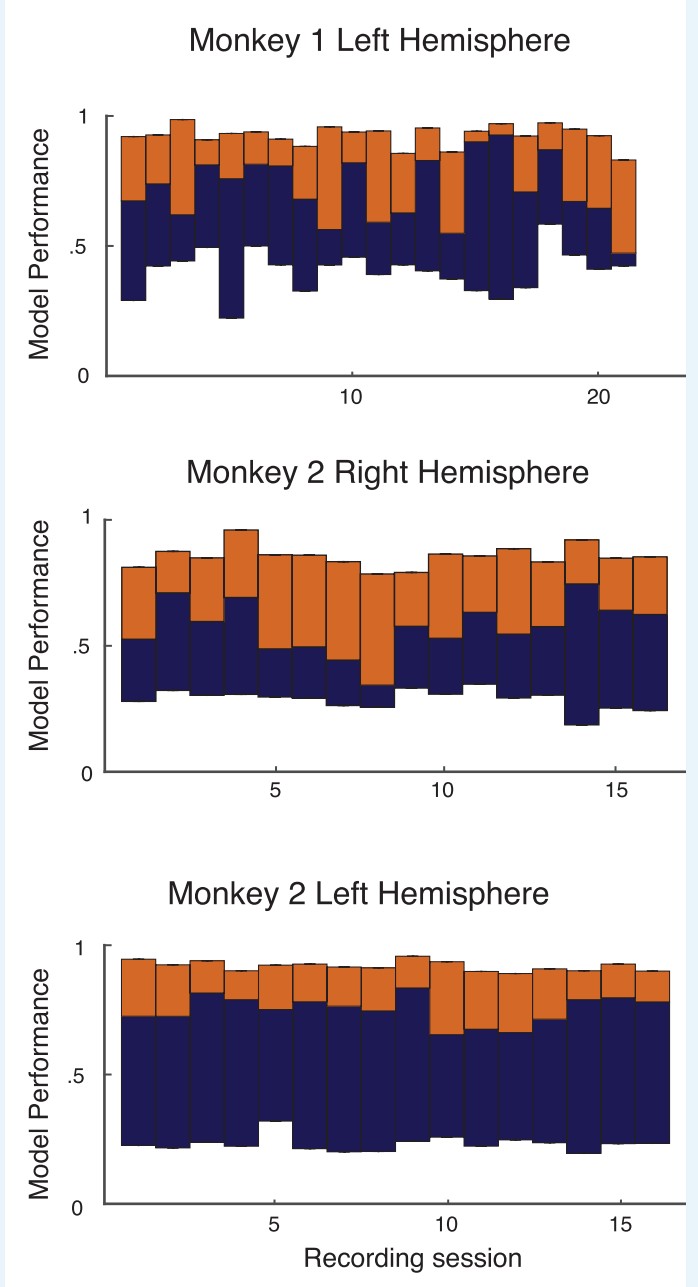

**Appendix 1—figure 3.** Performance of basic analysis on model on individual recording sessions from left hemisphere of monkey 1, and both hemispheres of monkey 2. The format and colors match that described in *Figure 2* of the main text.

## Low-dimensional modulation is intrinsic to neurons

In order to further test our model, we asked to what extent the actual value of the covariance gain $g_i$ of neuron $i$ depends on the neural population whose covariance matrix $g_i$ was estimated from. If we had solved the system $S$ of equations $C_{i,j}^A = g_i g_j C_{i,j}^U$ using covariance matrices computed from recordings from a different set of neurons (including neuron $i$), would the value of $g_i$ be different? If not, this would be further indication of the independence of the attentional modulation of neuron $i$ from the particular set of other neurons it is analyzed with.

We tackle this question by dividing a set of $N$ neurons into $k$ sets $S_i^{(1)}, S_i^{(2)}, ..., S_i^{(k)}$ of $m \equiv (N+1)/2$ neurons each that all contain the neuron $n_i$ ($m \equiv N/2 + 1$ if $N$ is originally even). As an example take $k = 2$ and consider the set of neurons $n_1, ..., n_{2i-1}$ partitioned into two subsets $S_i^{(1)} = \{n_1, ..., n_i\}$ and $S_i^{(2)} = \{n_i, ..., n_{2i-1}\}$ (**Appendix 1—figure 5a**). We solve **Equation (1)** using the systems of equations obtained from $S_i^{(1)}$ and $S_i^{(2)}$, and obtain two solutions $\mathbf{g}_i^{(1)}$ and $\mathbf{g}_i^{(2)}$. We take the variance of the $g$-estimations as a metric for how closely the different subsets can estimate an intrinsic value of $g$. A higher variance would indicate a poorer convergence, and therefore a lower degree of independence from other neurons. **Appendix 1—figure 5b** shows the spread of $g$-estimates from one dataset for the data, as well as the upper (UB) and lower (shuf) bounds. This spread includes estimates for all $g$-values for all neurons. The spread in the shuffled case (SEM= 7.42) is largest by two orders of magnitude, and the spread of the upper bound (SEM= $2.60 \times 10^{-3}$) is only one order of magnitude tighter than that of the data (SEM= $1.03 \times 10^{-2}$), so this case is close to ideal.

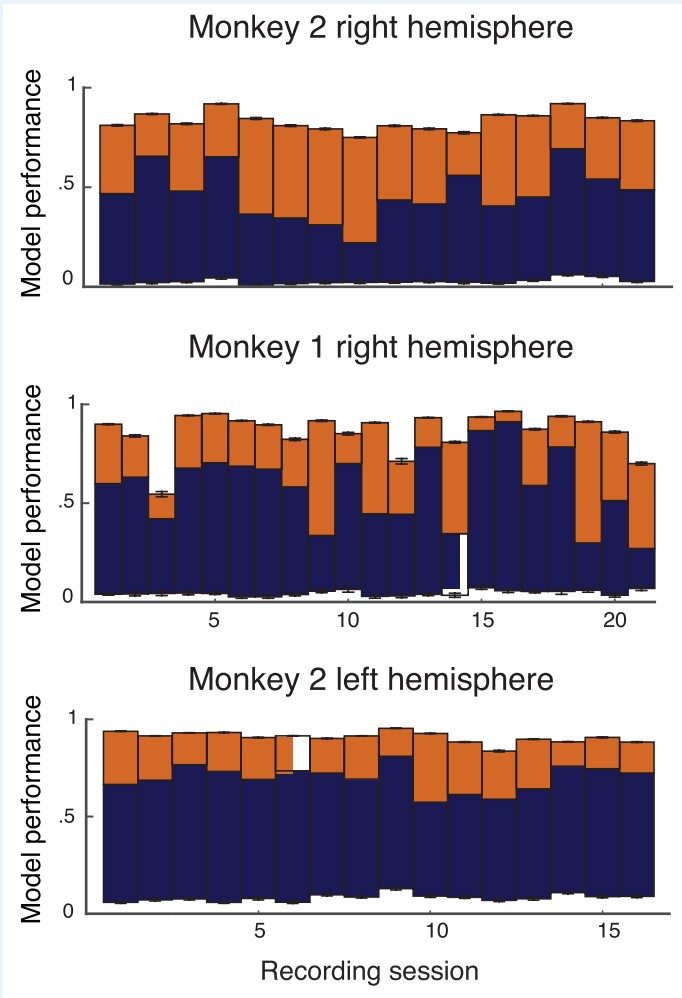

**Appendix 1—figure 4.** Performance of leave-one-out cross-validation on model on data from individual recording sessions from the left hemisphere of Monkey 1, and both hemispheres of Monkey 2. The format and colors match that described in **Figure 2** of the main text.

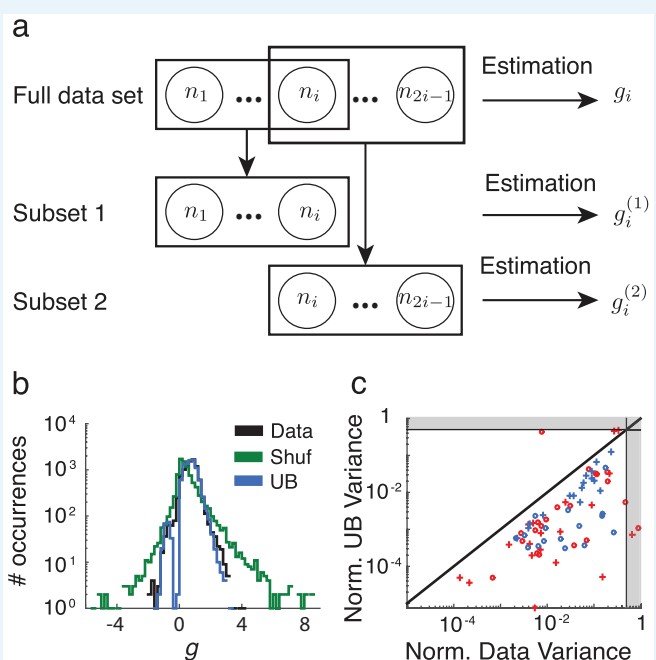

**Appendix 1—figure 5.** Overlap analysis of gain parameters. (**a**) Schematic of overlap analysis. A set of $n_{2i-1}$ neurons is divided into two sets $S_1$ and $S_2$ of $i$, which overlap by exactly one neuron, indexed without loss of generality as neuron $i$. Parameter $g_i$ is computed using $S_1$ and $S_2$, resulting in two estimates $g_i^{(1)}$ and $g_i^{(2)}$. (**b**) Spread of $g$ estimates for the data (black), as well as the upper (blue) and lower (green) bounds, from one day of recordings in one monkey. (**c**) Mean variance of the $g$ estimates computed from the data (abscissa) vs from the upper bound (ordinate), normalized by the mean shuffled variance. Each color denotes one of the monkeys, circles denote the right hemisphere recordings, and plusses denote the left hemisphere recordings. The gray regions consist of those points that are beyond 0.5, and therefore closer to the lower bound than the upper bound.

For each data set, the analysis is done for each neuron for 100 different permutations of the neurons to generate $S_i^{(k)}$, $k = 1, ..., 100$. For shuffled and upper-bound analysis, 10 shuffles or Poisson realizations, and 10 permutations were used. In all cases there was a total of $100 \times$ #neurons points. **Appendix 1—figure 5c** shows an overview of the performance for all datasets. The abscissa is the mean variance of the $g$-estimates computed from the data, normalized by the mean variance computed from the shuffled data: $\frac{\langle \text{Var}_k(g_i^{(k)}) \rangle_i}{\langle \langle \text{Var}_k(g_{i,\text{shuf}}^{(k)}) \rangle_{\text{shuf}} \rangle_i}$. The 'shuf' subscript denotes averaging over each shuffle. The ordinate is the mean variance of the $g$-estimates computed from the upper bound, with the same normalization: $\frac{\langle \langle \text{Var}_k(g_{i,UB}^{(k)}) \rangle_{\text{poiss}} \rangle_i}{\langle \langle \text{Var}_k(g_{i,\text{shuf}}^{(k)}) \rangle_{\text{shuf}} \rangle_i}$. The 'poiss' subscript denotes averaging over each Poisson realization of the upper bound covariance matrix. We chose to normalize the mean data and upper-bound variances by the mean shuffled variance so that a value of 1 would mean equality to the lower bound, meaning the only detected structure comes from finite-size effects, and a value of 0 would mean perfect convergence of the $g$-estimates. The gray regions are a visualization for the points which are closer to 1 than 0 (values above 0.5) on the log-axes. Most of the data unsurprisingly falls below the diagonal, so the variance is greater for the data than the upper bound. Less trivially, most of the data falls outside of the gray regions, and are much closer to 0 than 1, indicating excellent performance. This implies a structure in the modulation of the (unshuffled) covariance matrices that is preserved over analysis in the contexts of different groups of other neurons. In other words, attention modulates the individual neurons to a large extent independently, in a low-dimensional manner.

# Network requirements for attentional modulation

In this section we study a network of $N$ neurons with the spike train output from neuron $i$ being $y_i(t) = \sum_k \delta(t - t_{ik})$ where $t_{ik}$ is the $k^{th}$ spike time from neuron $i$. We consider multiple trials of the discrimination experiment and model the spike train only over a time period $t \in (0, T)$, where we assume that the spike trains to have have reached equilibrium statistics. We abuse notation and take the spike count from neuron $i$ over a trial as $y_i = \int_0^T y_i(t)dt$. The trial-to-trial covariance matrix of the network response is $\mathbf{C}$ with element $c_{ij} = \mathrm{Cov}(y_i, y_j)$.

To analyze the network activity we first assume that each spike train is simply perturbed about a background state and employ the linear response ansatz (*Ginzburg and Sompolinsky, 1994*; *Doiron et al., 2004*; *Trousdale et al., 2012*) :

$$y_i = y_{iB} + L_i \left( \sum_{k=1}^{N} J_{ik} y_k + \xi_i \right). \tag{28}$$

Here, $J_{ik}$ is the synaptic coupling from neuron $k$ to neuron $i$ (proportional to the synaptic weight), and $\xi_i$ is a fluctuating external input given to neuron $i$. The background state of neuron $i$ is $y_{iB}$, and it represents the stochastic output of a neuron that is not due to the recurrence from the network ($J = 0$) or the external input ($\xi_i = 0$). Finally, $L_i$ is the input to output gain of a neuron $i$. In this framework, $y_i$, $y_{iB}$, and $\xi_i$ are random variables, while $L_i$ and $J_{ik}$ are parameters that describe the intrinsic and network properties of the system. Without loss of generality we take $\langle y_{iB} \rangle = 0$, $\langle \xi_i \rangle = 0$, making $\langle y_i \rangle = 0$ a solution for the mean activity. We remark that formally *Equation (28)* is incorrect as written; $y_i$ is a random integer while, for instance, $L_i J_{ik} y_k$ need not be an integer. *Equation (28)* is only correct upon taking an expectation (over trials) of $y_i$.

Here we derive the requirements for external fluctuations and internal coupling for network covariability $\mathbf{C}$ to satisfy the following two conditions (on average):

**C1:** $c_{ij}^A = g_i g_j c_{ij}^U$ ; attentional modulation of covariance is rank one.

**C2:** $g_i < 1$ ; spike count covariance decreases with attention.

It is convenient to write *Equation (28)* in matrix form and isolate for the population response:

$$\vec{\mathbf{y}} = (\mathbf{I} - \mathbf{K})^{-1} \left( \vec{\mathbf{y}}_\mathbf{B} + \mathbf{L}\vec{\xi} \right). \tag{29}$$

Here $\vec{\mathbf{y}} = [y_1, \dots y_N]^T$ with similar notation for $\vec{\mathbf{y}}_\mathbf{B}$ and $\vec{\xi}$. The matrix $\mathbf{K}$ has element $\mathbf{K}_{ij} = L_i J_{ij}$, while $\mathbf{L} = \mathrm{diag}(L_i)$ and $\mathbf{I}$ is the identity matrix. Using *Equation (29)* we can express the covariance matrix $\mathbf{C} = \langle \vec{\mathbf{y}}\vec{\mathbf{y}}^T \rangle$ as:

$$\mathbf{C} = \underbrace{(\mathbf{I} - \mathbf{K})^{-1} \mathbf{B} (\mathbf{I} - \mathbf{K}^T)^{-1}}_{\text{internal covariability}} + \underbrace{(\mathbf{I} - \mathbf{K})^{-1} \mathbf{L}\mathbf{X}\mathbf{L} (\mathbf{I} - \mathbf{K}^T)^{-1}}_{\text{external covariability}}, \tag{30}$$

where $^T$ denotes the transpose operation. Here $\mathbf{B} = \langle \vec{\mathbf{y}}_\mathbf{B}\vec{\mathbf{y}}_\mathbf{B}^T \rangle$ is the background covariance, which we take to be simply $\mathbf{B} = \mathrm{diag}(b_i)$. The input covariance matrix is $\mathbf{X} = \langle \vec{\xi}\vec{\xi}^T \rangle$ with elements $x_{ij}$. In the above we assumed that $\langle \vec{\mathbf{y}}_\mathbf{B}\vec{\xi}^T \rangle = \mathbf{0}$, meaning that the background state is uncorrelated with the external noisy input.

It is clear that $\mathbf{C}$ naturally decomposes into two terms. The first term represents the correlations that are internally generated within the network, via the direct synaptic coupling

$\mathbf{K}$ acting upon the background state $\mathbf{B}$. The second term is how the direct synaptic coupling $\mathbf{K}$ filters the externally applied correlations $\mathbf{X}$.

## Satisfying C1

The background matrix $\mathbf{B}$ is a diagonal matrix and is hence rank $N$. The high rank $\mathbf{B}$ combined with attentional modulations of both $\mathbf{B}$ and $\mathbf{K}$ make it impossible to satisfy condition **C1**. If the spectral radius of $\mathbf{K}$ is less than 1, then we can expand $(\mathbf{I} - \mathbf{K})^{-1} = \mathbf{I} + \sum_{n=1}^{\infty} \mathbf{K}^n$ (**Pernice et al., 2011**; **Trousdale et al., 2012**). Inserting this expansion into the expression for the internally generated covariability yields:

$$(\mathbf{I} - \mathbf{K})^{-1}\mathbf{B}(\mathbf{I} - \mathbf{K}^T)^{-1} = \mathbf{B} + \mathbf{B}\mathbf{K}^T + \mathbf{K}\mathbf{B} + \mathbf{K}\mathbf{B}\mathbf{K}^T + \cdots.$$

Extracting the covariance between neuron $i$ and $j$ ($i \neq j$) due to internal coupling within the network gives:

$$c_{ijB} = b_i L_j J_{ji} + b_j L_i J_{ij} + \sum_k L_i L_j b_k J_{ik} J_{jk} + \cdots.$$

If we take $J_{ij} \sim 1/N$ and the network connectivity to be dense (meaning the connection probability is $\sim \mathcal{O}(1)$) then each term is $\mathcal{O}(1/N)$. So long as the spectral radius of $\mathbf{K}$ is less than one then the series converges and as $N \to \infty$ we have that $c_{ijB}$ vanishes (**Pernice et al., 2011**; **Trousdale et al., 2012**; **Helias et al., 2014**).

This argument can be extended to networks with $J_{ij} \sim 1/\sqrt{N}$ when combined with a balance condition between recurrent excitation and inhibition. Such networks also produce an asynchronous state where $c_{ijB} \sim 1/N$, vanishing in the large $N$ limit (**Renart et al., 2010**). However, formally balanced networks in the asynchronous state with $N \to \infty$ have solutions that do not depend on the firing rate transfer $L$. The attention dependent modulation $A_L : L^U \to L^A$ is a critical component of our model and care must be taken in ensuring that

In contrast, the external covariance $\mathbf{X}$ is not a diagonal matrix, so that the contributions from external fluctuations to $\mathbf{C}$ scale as $N^2 J^2$. This is $\mathcal{O}(1)$ for $J \propto 1/N$. Thus, while the terms in $\mathbf{X}$ must be weak for the linear approximation in **Equation (28)** to hold, they need not vanish for large $N$. Indeed, for moderate $\mathbf{X}$ and large network size it is reasonable to ignore the contribution of internally generated fluctuations to $\mathbf{C}$. Recent analysis of cortical population recordings show that the shared spiking variability across the population can be well approximated by a rank one model of covariability (**Ecker et al., 2014**; **Lin et al., 2015**; **Ecker et al., 2015**; **Rabinowitz et al., 2015**). Thus motivated, we take the external fluctuations $\mathbf{X} = \mathbf{x}\mathbf{x}^T$ where $\mathbf{x} = [x_1, \ldots, x_N]^T$. In total, we have for large $N$ the approximation:

$$\mathbf{C} \approx \left((\mathbf{I} - \mathbf{K})^{-1}\mathbf{L}\mathbf{x}\right)\left((\mathbf{I} - \mathbf{K})^{-1}\mathbf{L}\mathbf{x}\right)^T = \mathbf{c}\mathbf{c}^T. \tag{31}$$

Hence $\mathbf{C}$ is rank one matrix with $\mathbf{c} = \left((\mathbf{I} - \mathbf{K})^{-1}\mathbf{L}\mathbf{x}\right) = [c_1, \ldots, c_N]^T$. It is trivial to satisfy condition **C1** with $g_i = c_i^A / c_i^U$.

## Satisfying C2

We again use the expansion $(\mathbf{I} - \mathbf{K})^{-1} = \mathbf{I} + \sum_{n=1}^{\infty} \mathbf{K}^n$. Truncating this expansion at $n = 1$ yields an approximation considering only synaptic paths of length one in the network, and neglecting higher order paths. This is appropriate for $J_{ij}$ sufficiently small. Truncating after inserting the expansion into **Equation (30)** yields the following approximation for $\mathbf{c}$:

$$\mathbf{c} \approx (\mathbf{I} + \mathbf{K})\mathbf{L}\mathbf{x}. \qquad (32)$$

The analysis in the main text begins with this approximation to derive *Equation (5)* of the main text.

## Spiking network

### Spiking network description

We implement a network of leaky integrate-and-fire neurons (LIF) with 1000 excitatory neurons and 200 inhibitory neurons. Individual neurons were modeled as integrate-and-fire units whose voltages obeyed

$$\frac{dV_i}{dt} = \frac{1}{\tau}(\mu_i - V_i) + I_i^{\text{syn}} + I_i^{\text{ext}} \qquad (33)$$

for neuron $i$. When the voltage reached a threshold $V_{\text{th}} = 1$, a spike was recorded and the voltage reset to $V_{\text{re}} = 0$. Time was measured in units of the membrane time constant, $\tau = 1$ for all neurons. The bias $\mu$ depended on neuron type and attentional state. In the unattended state, the bias for excitatory neurons was $\mu_E^{\text{un}} = 0.6089$ and $\mu_I^{\text{un}} = 0.5388$. In the attended state, $\mu_E^{\text{att}} = 0.8713$ and $\mu_I^{\text{att}} = 0.8996$. The recurrent input to neuron $i$ was

$$I_i^{\text{syn}}(t) = \sum_j \mathbf{W}_{ij}\mathbf{J}_{ij}(t) * y_j(t) \qquad (34)$$

where $\mathbf{W_{ij}}$ is the strength of the connection from neuron $j$ to neuron $i$, $J_{ij}(t)$ is the synaptic filter for the projection from neuron $j$ to neuron $i$, $*$ denotes convolution and $y_j(t)$ is neuron $j$'s spike train – a series of $\delta$-functions centered at spike times. The synaptic filters were taken to be alpha functions,

$$J_{ij}(t) = \frac{t}{\tau_s}e^{-t/\tau_s} \qquad (35)$$

with $\tau_s = 0.3$ of the passive membrane time constant for all synapses. The connection probability from neurons in population $A$ to population $B$ was $p^{AB}$, with $p^{EE} = 0.2$ and $p^{EI} = p^{IE} = p^{II} = 0.4$. Synaptic weights for connections between excitatory neurons were $\mathbf{W}^{EE} = 0.0075$ and $\mathbf{W}^{IE} = 0.0037$, $\mathbf{W}^{EI} = -0.0375$, $\mathbf{W}^{II} = -0.0375$. These parameters, and the bias voltages $\mu$, were chosen so that the mean field theory derived above was valid for the spiking network's firing rates.

The excitatory neurons were divided into four clusters, each excitatory neuron receiving half of its inputs from neurons in the same cluster and half from others. Projections to and from inhibitory neurons were unclustered.

External input from outside the network was contained in $I_i^{\text{ext}}$. We modeled this as a partially correlated Gaussian white noise process: $I_i^{\text{ext}}(t) = \sigma_i\big(\sqrt{1-c}\xi_i(t) + \sqrt{c}\xi_c(t)\big)$. $\xi_i(t)$ was Gaussian white noise private to neuron $i$ and $\xi_c(t)$ was shared between all neurons. $c = 0.05$ denoted the fraction of common input and the noise intensity for excitatory neurons was $\sigma_E = 0.3$ and for inhibitory neurons $\sigma_I = 0.35$.

The firing rate of neuron $i$ in a trial of length $L$ is given by its spike count in that trial $n_i^L$, $r_i = \langle n_i^L \rangle / L$ where $\langle \cdot \rangle$ denotes averaging over trials. The spike train covariance between neurons $i$ and $j$ describes the above-change likelihood that action potentials occur in each spike train separated by a time lag $s$:

$$q_{ij}(s) = \frac{1}{L}\int_0^L \langle y_i(t)y_j(t-s)\rangle dt - r_i r_j. \tag{36}$$

For simulations, we measure the population-averaged spike train cross-covariance function $Q(s) = (N_E(N_E - 1))^{-1}\sum_{i,j=1,i\neq j}^{N_E} q_{ij}(s)$ by average a randomly chosen subsample of 100 spike train cross-covariances from pairs of neurons in the same cluster.

In order to calculate the covariance of neuron $i$ and $j$'s spike counts in windows of length $T$, $n_i^T$ and $n_j^T$, we use the relation

$$\begin{aligned}
\mathrm{Cov}\left(n_i^T, n_j^T\right) &\equiv \langle n_i^T n_j^T\rangle - \langle n_i^T\rangle\langle n_j^T\rangle \\
&= \int_{-T}^{T} q_{ij}(s)(T - |s|)
\end{aligned} \tag{37}$$

The cross-correlation of input currents was averaged over the same random subsample of the network as the spike train covariances. Current cross-correlations were normalized so that each current's autocorrelation at zero lag was 1.

## Spiking network analysis

The LIF model simulates voltages and produces spike trains, from which we can compute firing rates and covariances. *Appendix 1—figure 6a,b* show example voltage traces of individual excitatory neurons, with the spikes they produce shown above. Note that in the attended state, more spikes are produced, corresponding to a higher firing rate. *Appendix 1—figure 6c,d* show rasters for all the neurons in the unattended (c) and attended (d) states. Higher firing rates can be observed, especially for the inhibitory neurons. Averaging the spike trains over the excitatory population gives us the PSTH of the excitatory neurons. *Appendix 1— figure 6e*, left shows the unattended (turquoise) and attended (orange) PSTH smoothed with a sliding Gaussian window with width (std dev) 10 ms. The histograms on the right demonstrate the decrease in population variance with attention.

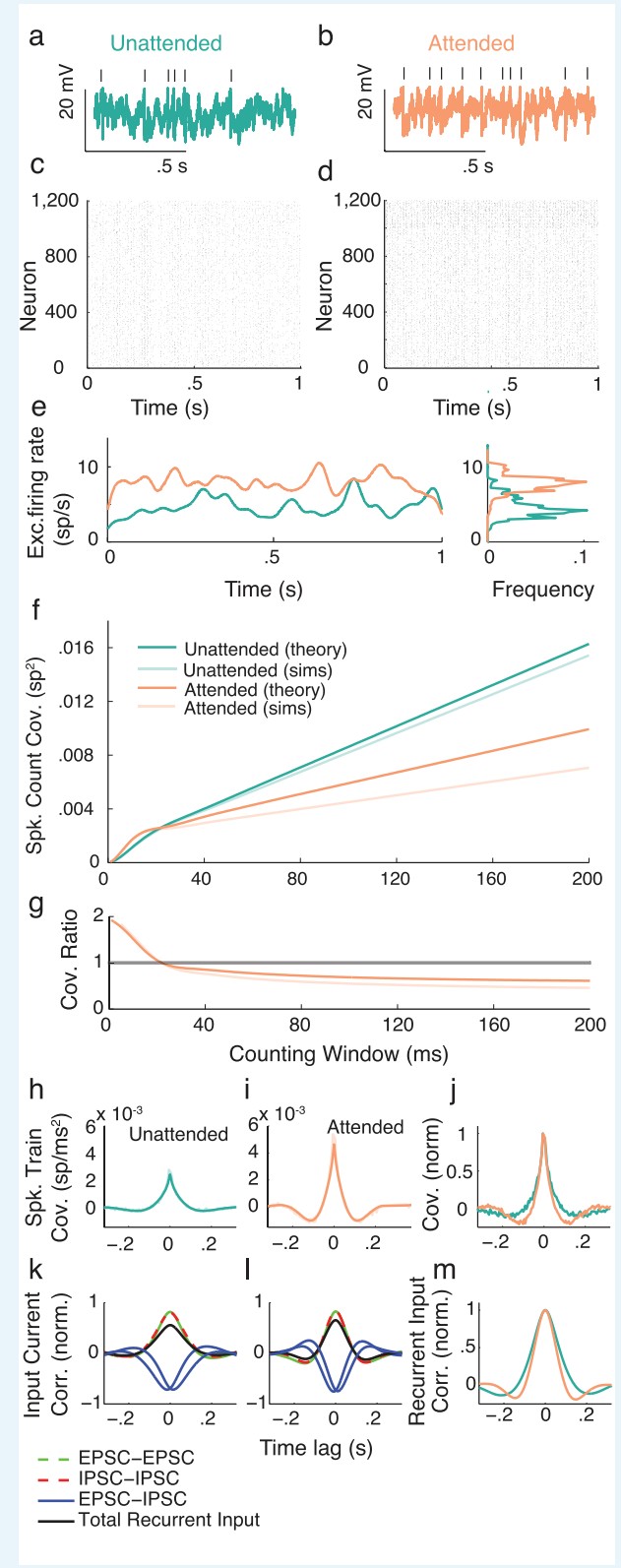

**Appendix 1—figure 6.** Spiking model and simulations. (**a,b**) Example voltage trace from an excitatory model neuron in the unattended (a) and attended (b) states. Top tick marks denote spike times. (**c**) Raster plot of neurons in the unattended state. Neurons 1 to 1000 are excitatory, and 1001 to 1200 are inhibitory. (**d**) Raster plot of neurons in the attended state. (**e**) Excitatory population-averaged firing rates for the unattended (turquoise) and

attended (orange) states. Right: frequency distributions of population-averaged firing rates. (**f**) Mean pairwise spike count covariance for different counting windows. Other than an increase in synchrony on very small timescales due to gamma oscillations, the spike count covariance decreases with attention regardless of counting window. (**g**) Ratio $R_{Cov}$ of attended and unattended spike count covariance, as a function of counting window. (**h, i**) Derived (solid turquoise) and simulated (muted turquoise) spike train cross-covariance functions of excitatory neurons in the unattended (h) and attended (i) states, averaged over pairs. (**j**) Spike train cross-covariance functions of excitatory neurons in the unattended and attended states, normalized to peak at 1. (**k, l**) Normalized input current cross-correlation functions of excitatory inputs to pairs of neurons (dashed green), inhibitory inputs to pairs of neurons (dashed red), excitatory and inhibitory inputs to pairs of neurons (blue), and summed excitatory and inhibitory recurrent inputs to pairs of neurons (black), in the unattended (k) and attended (l) states. (**m**) Attended (orange) vs unattended (turquoise) recurrent input cross-correlation functions. The excitatory cross-correlation function is narrower, just as for the output cross-covariance function, so the effects are happening on the level of inputs.

The spiking model provides the opportunity to directly compute the pairwise spiking covariance, in addition to the population variance. **Appendix 1—figure 6f** shows the pairwise spike count covariance computed over counting windows from 0 to 200 ms. For small counting windows, corresponding to high-frequency correlations, neurons in the attended state have slightly higher spike count covariance. This is consistent with the slightly higher peak in the attended autocovariance function from the mean-field theory (**Figure 4e**, main text), as well as experimental results (**Fries et al., 2001**). For counting windows greater than 30 ms, the spike count covariance notably decreases with attention. The experiments we are modeling (**Cohen and Maunsell, 2009**) measure spike count correlations over 200 ms counting windows, corresponding to the right-most points in **Appendix 1—figure 6f**. The proportional changes in the spike count covariance are expressed in the covariance ratio $R_{\mathrm{Cov}} = \mathrm{Cov}^A(n_1, n_2)/\mathrm{Cov}^U(n_1, n_2)$, shown in **Appendix 1—figure 6g**. Values of $R_{\mathrm{Cov}}$ greater than one indicate increased spike count covariance with attention, and values of $R_{\mathrm{Cov}}$ less than one indicate decreased spike count covariance with attention. The crossing of the $R_{\mathrm{Cov}} = 1$ line is apparent at counting windows of approximately 30 ms. The theoretical values were computed using linear response theory (**Trousdale et al., 2012**).

To dissect the spike count covariance by different time lags, we consider the spike train covariance function (**Equation 36**), which is the pairwise-neuron analogue of the autocovariance function of the population-averaged activity (**Figure 5e**, main text). **Appendix 1—figure 6h,i** show the spike train covariance functions of excitatory neurons in the unattended and attended states. To compare the two, **Appendix 1—figure 6j** shows them normalized so that their maximum values are 1. In accordance with our mean-field results, the attended spike train covariance decays faster than the unattended spike train covariance, indicating increased stability in the attended state.

The spiking model also provides the opportunity to investigate the inputs to individual neurons, something that is difficult to do experimentally, and does not apply to mean-field models. **Appendix 1—figure 6k,l** shows the correlation functions of different types of inputs to a pair of excitatory neurons, averaged over pairs of excitatory neurons, in the unattended (k) and attended (l) states. Computing the correlation functions of the total recurrent input (black curves) reveals that correlations between excitatory inputs (EPSC-EPSC, dashed green), and correlations between inhibitory inputs (IPSC-IPSC, dashed red), are canceled by anti-correlations between excitatory and inhibitory inputs (EPSC-IPSC, blue). This is consistent with the idea of correlation cancellation by inhibitory tracking of excitatory activity (**Renart et al., 2010**; **Tetzlaff et al., 2012**; **Ly et al., 2012**). Attention, by shifting the system into a more stable state, allows this cancellation to occur more efficiently, thereby reducing the pairwise covariance. **Appendix 1—figure 6m** shows the input current correlation functions of the total recurrent inputs to pairs of excitatory neurons, normalized to peak at 1.

We conclude that the correlation cancellation brought about by recurrent inhibitory feedback suppresses correlations of the total recurrent input, which in turn decreases the output correlations.

