## [Decision Letter]

Thank you for submitting your article "Attentional modulation of neuronal variability in circuit models of cortex" for consideration by *eLife*. Your article has been reviewed by three peer reviewers, one of whom is a member of our Board of Reviewing Editors and the evaluation has been overseen by Timothy Behrens as the Senior Editor. The following individual involved in review of your submission has agreed to reveal his identity: Ruben Coen-Cagli (Reviewer #3).

The reviewers have discussed the reviews with one another and the Reviewing Editor has drafted this decision to help you prepare a revised submission.

All three reviewers found the paper interesting and potentially important. Of particular note is the demonstration that a single source of attentional modulation in an E-I network can explain both increases in firing rate and decreases in noise covariance, without the need to postulate separate effects. In addition, the modelling is solid and convincing, and the result is an important advance in our understanding of attentional effects.

However, there are a couple if issues that need to be addressed:

1) Robustness to network parameters and assumptions needs to be explored.

2) The rank-1 covariance needs to be better quantified.

3) Information was about contrast, whereas the experiment was about orientation change detection. This is important because the rank 1 covariance matrix is unlikely to have much effect on information about orientation.

It won't be completely trivial to address these issues, but we believe they are addressable.

It is *eLife*'s policy to provide a summary of essential revisions. That's hard to do for these reviews, as they were all relatively extensive. I am, though, going to give it a shot. The exposition will be a little uneven, as I combined the reviews, without trying to edit for uniformity.

1) A big issue is robustness to parameters. Essentially what we want to know is: what are the constraints on parameter space for the results to hold qualitatively? It's probably hard to fully answer this, but it should be possible to provide answers for some of the more important parameters. Following are some more specific points.

First is a long comment about two of the modelling assumptions:

- The weights scale as 1/*N*.

- Perfect balance (J*_EE_* = J*_IE_* = J*_E_* and J*_II_*= J*_EI_* = J*_I_*).

This 1/*N* scaling is different from the usual one, which is 1Nand in that regime perfect balance is problematic. Granted, the 1Nscaling is probably too large, but 1/*N* is probably too small. At the very least, the authors need to comment on this scaling – after all, the last author just published a paper on correlations which was based on 1Nscaling. We're not suggesting that the analysis be redone, but it would be good to know whether the analysis really does apply to biologically plausible connectivity.

Now, a couple of technical comments relating to these points.

The first one is mainly a suggestion. The authors use the 1/*N* scaling to argue that the first term in Equation 3 of SM (the full-rank component of the covariance matrix) is O(1/*N*). But this may not be necessary. An instructive case is completely homogeneous coupling, in which J*_ij_*depends only on the type (*E* versus *I*) of neurons *_i_*and *_j_*, the probability of connection is 1. In this case, if the scaling is 1Na back of the envelope calculations indicates that the first term in Equation 3 of SM scales as 1/*N*. I believe that if iid noise is added to the weight matrix (while retaining the 1Nscaling), the first term in Equation 3 of SM would still scale as 1/*N*, but I'm not sure. This should probably be checked: if it one turns out to be correct, it would go a long way toward dispelling doubts about the 1/*N* scaling.

Second, in Equation 7 the authors derive an expression for the variance over long time windows. This was derived under the perfect balance assumption. If that assumption is dropped, there's an additional term in the denominator that scales as L*_E_* L*_I_ -Det(J)* (where *J* is the 2x2 matrix of weights). If the components of *J* are large – as they probably are in realistic networks -- this can have a large effect. Is it possible to estimate its effect for realistic networks? How would that change the results?

A couple semi-minor points on robustness:

a) Equation 7 depends on σ*_E_* – σ*_I_*. That's taken to be negative (Table 1). How much do the results change if it's positive?

b) In real networks, an increase in drive (which is how top down attention is modelled) would probably lead to an increase in noise (because variance scales with spike count). I think it would be important to estimate how large an increase in noise could be tolerated without an increase in covariance with attention.

2) The rank-1 covariance

a) The authors tell us that the modulation matrix is close to being rank one, but they tell us nothing about the vector of modulation gains defining this rank one matrix. I would like to see answers to basic questions such as: What is the distribution of *gs*? Are they correlated with as (firing rate modulations) and/or with baseline firing rates? Or with the vectors obtained using low-rank approximation of the covariance matrix itself? The model must make specific predictions about the answers to all these questions, and it would be nice to see these predictions tested.

b) It is unclear why the authors used their method for low rank approximation, as opposed to more standard methods based on SVD (that naturally provides a quantification of the quality of a general low rank approximation based on singular values). I think it would be useful to check what they get using alternate methods, to check the robustness of their results.

c) The data in Figure 1 show a broad range of effects of attention on noise covariance, but the model addresses only the overall reduction in the mean, not any other property of the distribution (including the fact that there are a substantial minority of cases with increased covariance under attention). Isn't is possible to study the distribution across the network model, at least in simulations? And again, the structure of the covariance (and tuning) is important to determine information.

d) A separate, smaller issue is the assumption that attention acts as a low-rank modulation of noise covariance. The opening statement in the Results, subsection “Attention as a low-rank modulation of noise covariance**”** is that "we need to first understand the dimension of attentional modulation", as if a model-comparison of some sort was going to be performed between low-rank and full-rank modulations. Instead, there is only a quantification that the low-rank assumption works reasonably well, but no comparison to a higher-rank description. Also, why is the assumption of a multiplicative effect better than e.g. additive modulation? This could be quantified.

3) Fisher information: contrast versus orientation.

My main concern is that I see a disconnect between the modeling and the data/experimental paradigm that motivate the modeling.

I am not convinced about the generality of the conclusions on the effects of attentional modulation on population coding in the model. The experiment is about orientation-change detection, but in the modeling the stimulus dependence is more like contrast (all neurons are identically modulated by the stimulus intensity) than like orientation (where a change in stimulus value would drive some neurons up, and others down). This is acknowledged in the closing paragraphs, and suggested as future work, but I wonder if it should instead be done as part of this paper. I am no expert in EI networks, I don't know how long it would take, but here is concretely what I would like to see and why. Add the stimulus drive in the actual network, not just in the mean field solutions. And while doing that, assume heterogeneity (and possibly nonlinearity) of tuning. If the stimulus acts like contrast, then doing the information analysis on the mean field is fine; but otherwise the mean field solution is effectively a suboptimal decoder (weight all neurons equally), and the conclusions about information may be only valid for that decoder, not for the optimal decoder. The rank-one external noise by itself does not limit information for orientation-like stimulus dimensions (unless you modify it to exactly align it with the signal) (e.g. Moreno-Bote et al., 2014), so some other source of differential correlations needs to be considered if you want the attentional modulation to have any chance of improving information.

We'll admit that this may be a hard one to address rigorously. But the authors should provide an extended discussion of this issue. And an attempt should be made to provide approximate calculations and/or estimates.

---

## [Author Response]

All three reviewers found the paper interesting and potentially important. Of particular note is the demonstration that a single source of attentional modulation in an E-I network can explain both increases in firing rate and decreases in noise covariance, without the need to postulate separate effects. In addition, the modelling is solid and convincing, and the result is an important advance in our understanding of attentional effects.

*However, there are a couple if issues that need to be addressed:*

We thank the reviewers for their very careful read of our manuscript and for their insightful comments concerning our work. We have made changes to the paper in response to these comments, and as a result we feel our manuscript has significantly improved. In particular, we have added two new figures (Figure 3 and Figure 6). We have also incorporated the previous supplementary section as an appendix, in response to both reviewer requests as well as those from the editorial staff. Below, we list our responses to each detailed comment from the reviewers and summarize the changes we have made.

*1) Robustness to network parameters and assumptions needs to be explored.*

*2) The rank-1 covariance needs to be better quantified.*

*3) Information was about contrast, whereas the experiment was about orientation change detection. This is important because the rank 1 covariance matrix is unlikely to have much effect on information about orientation.*

*It won't be completely trivial to address these issues, but we believe they are addressable.*

*It is eLife's policy to provide a summary of essential revisions. That's hard to do for these reviews, as they were all relatively extensive. I am, though, going to give it a shot. The exposition will be a little uneven, as I combined the reviews, without trying to edit for uniformity.*

*1) A big issue is robustness to parameters. Essentially what we want to know is: what are the constraints on parameter space for the results to hold qualitatively? It's probably hard to fully answer this, but it should be possible to provide answers for some of the more important parameters. Following are some more specific points.*

*First is a long comment about two of the modelling assumptions:*

*- The weights scale as 1/N.*

*- Perfect balance (J_EE_ = J_IE_ = J_E_ and J_II_ = J_EI_ = J_I_).*

The reviewers raise two excellent points and we outline our response to both of them below.

*This 1/N scaling is different from the usual one, which is*
1/N
*and in that regime perfect balance is problematic. Granted, the*
1/N*scaling is probably too large, but 1/N is probably too small. At the very least, the authors need to comment on this scaling – after all, the last author just published a paper on correlations which was based on*
1/N*scaling. We're not suggesting that the analysis be redone, but it would be good to know whether the analysis really does apply to biologically plausible connectivity.*

We understand the reviewer’s suggestion and also think that we should discuss the case where weights scale as 1/N. Our analysis aims to establish the network requirements for:

C1:Aij= gigi cUij; attentional modulation of covariance is rank one (Figure 2).

C2: g_i_ < 1: spike count covariance decreases with attention (Figure 1).

We will argue below that 1/Ncoupling will not affect how C1 is satisfied, but will technically prevent C2 (in the large N limit).

To establish C1 we first observe that the covariance C decomposes as:

Here the matrix K has element K_ij_ = *L_i_J_ij_* where *L_i_* is the linear response and *J_ij_* is the synaptic strength from neuron *j* to *i*. The issue at hand is that we require that the internally generated covariability vanish for large *N*. If the spectral radius of K is less than 1, then we can expand (I−K)−1=I+ ∑n=1∞Kn(Pernice et al., 2011; Trousdale et al., 2012). Inserting this expansion into the expression for the internally generated covariability yields:

(I − K)^−1^B(I − K^T)−1^ = B + BK^T^ + KB + KBK^T^ + · · ·.

Extracting the covariance between neuron *i* and *j (i* ≠ *j*) due to internal coupling within the network gives:

*c_ijB_ = b_i_L_j_J_ji_ + bjL_i_J_ij_ +*
∑KL*_i_L_j_b_k_J_ik_J_jk_ + · · ·.*

If we take *J_ij_* ∿ 1/*N* and the network connectivity to be dense (meaning the connection probability is ∿ O(1)) then each term is O (1/N). So long as the spectral radius of K is less than 1 then the series converges and as *N* → ∞ we have that *c_ijB_* vanishes (Pernice et al., 2011; Trousdale et al., 2012; Helias et al., 2014). As the reviewers intuit this argument can be extended to networks with *J_ij_*∼ 1/N, however the analysis is more complicated. When J ∼ 1/Nwe require a *balance condition* where large excitation and inhibition effectively cancel. Renart et al., 2010 showed that even densely coupled balanced networks produce an asynchronous state where *c_ijB_* ∿1/*N*, vanishing in the large *N* limit (Renart et al., 2010). Showing this requires that we consider the excitatory/inhibitory subnetworks and extend the balance condition so that *c_ijB_*is the sum of O(1) terms that nonetheless combine to be *O*(1/*N*).

While we appreciate the reviewers comment we feel that this is a somewhat tagential point and will derail the flow of the manuscript. The above arguments feature in our Supplementary Materials and in the main text we now write:

“Spiking covariability in recurrent networks can be due to internal interactions (through J*_ik_*) or external fluctuations (through ξ*_i_*), or both (Ocker, 2017). […] In these cases spiking covariability requires external fluctuations to be applied and subsequently filtered by the network.”

Condition **C2** involves the attentional modulation itself. In the main text we derived:

In the above each term involves the gain modulation *LA − LU;* recall that *L* = *dr/dµ* (slope of the firing rate curve). In networks with *J ∼*
1/Nthen for *N → ∞* the balance condition takes precedence and the firing rate solution does not depend upon the firing rate function *f* and consequently neither upon *L* (vanVreeswijk and Sompolinsky, 1988). As such a modulation of *L* will not change the ability of a balanced network to track an external input. Of course, finite size balanced networks may rescue us from this, or including short term plasticity mechanisms will also impart nonlinearities in the transfer (Mongillo et al., 2012). But we again feel that these issues are best left for another paper. We have addressed this in our resubmitted manuscript by including the sentences:

“In the above we considered weak synaptic connections where *J_ij_ ∼* 1*/N.* […] synaptic nonlinearities through short term plasticity (Mongillo et al., 2012) may be necessary to satisfy condition **C2** with large synapses.”

Now, a couple of technical comments relating to these points.

*The first one is mainly a suggestion. The authors use the 1/N scaling to argue that the first term in Equation 3 of SM (the full-rank component of the covariance matrix) is O(1/N). But this may not be necessary. An instructive case is completely homogeneous coupling, in which J_ij_ depends only on the type (E versus I) of neurons _i_ and _j_, the probability of connection is 1. In this case, if the scaling is*
1/N*), a back of the envelope calculations indicates that the first term in Equation 3 of SM scales as 1/N. I believe that if iid noise is added to the weight matrix (while retaining the*
1/N
*scaling), the first term in Equation 3 of SM would still scale as 1/N, but I'm not sure. This should probably be checked: if it one turns out to be correct, it would go a long way toward dispelling doubts about the 1/N scaling.*

We hope that the above discussion satisfies this point.

*Second, in Equation 7 the authors derive an expression for the variance over long time windows. This was derived under the perfect balance assumption. If that assumption is dropped, there's an additional term in the denominator that scales as L_E_ L_I_ Det(J) (where J is the 2x2 matrix of weights). If the components of J are large – as they probably are in realistic networks – this can have a large effect. Is it possible to estimate its effect for realistic networks? How would that change the results?*

This is an excellent point. The assumption of perfect balance was made so that the formula for population variance was compact (our old Equation 7). However, the reduction of population variance with our attentional modulation is not critically dependent on this assumption. To demonstrate this we have added the following new text to the manuscript as well as a new figure exploring the robustness of our result.

“The expression for *VE* given above (Equation 7) assumes a symmetry in the network coupling, namely that *J_EE_*= *J_IE_^≡^ J_E_*and *J_EI_*= *J_II_^≡^ J_I_.* […] In total, the inhibitory mechanism for attention mediated reduction in population variability is robust to changes in the recurrent coupling with the network.”

*A couple semi-minor points on robustness:*

*a) Equation 7 depends on σ_E_ – σ_I_. That's taken to be negative (Table 1). How much do the results change if it's positive?*

*b) In real networks, an increase in drive (which is how top down attention is modelled) would probably lead to an increase in noise (because variance scales with spike count). I think it would be important to estimate how large an increase in noise could be tolerated without an increase in covariance with attention.*

There are several parameters that we could vary, as well as the transfer functions *fE*and *fI* themselves. In most cases the parameters like *σ* are scaled by *J*. We hope that by changing the synaptic coupling *J* (see new Figure 6) that we have addressed these concerns somewhat. A full analysis remains to be done.

*2) The rank-1 covariance*

*a) The authors tell us that the modulation matrix is close to being rank one, but they tell us nothing about the vector of modulation gains defining this rank one matrix. I would like to see answers to basic questions such as: What is the distribution of gs? Are they correlated with as (firing rate modulations) and/or with baseline firing rates? Or with the vectors obtained using low-rank approximation of the covariance matrix itself? The model must make specific predictions about the answers to all these questions, and it would be nice to see these predictions tested.*

The reviewers raise some important points. To address them we have added some new analysis and a new figure to the manuscript.

“To further validate our model we show the distribution of *g_i_*s computed from the entire data set (Figure 3). […] This indicates that the circuit modulation of firing rates and covariance may not be trivially related to one another (Doiron et al., 2016).”

*b) It is unclear why the authors used their method for low rank approximation, as opposed to more standard methods based on SVD (that naturally provides a quantification of the quality of a general low rank approximation based on singular values). I think it would be useful to check what they get using alternate methods, to check the robustness of their results.*

The low rank approximation we require is that [**gg***T*]*ij* = [**C***U*]*ij /*[**C***A*]*ij* (here [**A**]*ij* denotes the *ij*th element of matrix **A**). The number of trials are limited in our data and while we are confident that **C** is well estimated the ratio matrix [**C***U*]*ij /*[**C***A*]*ij* unfortunately is not. The justification for the low rank approximation of *C* is given in Rabinowitz et al., 2016 as they analyze the exact same data that we have analyzed.

*c) The data in Figure 1 show a broad range of effects of attention on noise covariance, but the model addresses only the overall reduction in the mean, not any other property of the distribution (including the fact that there are a substantial minority of cases with increased covariance under attention). Isn't is possible to study the distribution across the network model, at least in simulations? And again, the structure of the covariance (and tuning) is important to determine information.*

Our study focuses on a mean field theory of population dynamics. Necessarily, this theory cannot address the heterogeneity of correlations in the data. Our spiking simulations do have heterogeneity, owing to the random coupling within the network. However, the simplified binary coupling makes this heterogeneity quite weak. A full accounting of the spread in noise correlations shown in Figure 1 would require a better understanding of the mechanistic source of the heterogeneity in the network firing rates and variability itself. We feel that this is beyond the scope of this study and hope that our field theory provides sufficient insight into the main mechanisms underlying attentional modulation.

*d) A separate, smaller issue is the assumption that attention acts as a low-rank modulation of noise covariance. The opening statement in the Results, subsection “Attention as a low-rank modulation of noise covariance**”** is that "we need to first understand the dimension of attentional modulation", as if a model-comparison of some sort was going to be performed between low-rank and full-rank modulations. Instead, there is only a quantification that the low-rank assumption works reasonably well, but no comparison to a higher-rank description. Also, why is the assumption of a multiplicative effect better than e.g. additive modulation? This could be quantified.*

A high rank model (rank *N*) of attentional modulation would always work perfectly since we would simply set *gij*= cijA/cijU. The problem is that each pair of neurons would require a modulation specific to them. From a biological standpoint this would be very complicated, while a rank 1 modulation is much simpler since only individual neurons need to be targeted. To express this thought we have reworded the text in the Results, subsection “Attention as a low-rank modulation of noise covariance” as follows (for reference Equation 1 is **C***A* = *AC ◦***C***U*):

“On the one hand *A_C_*could be constructed from the ratios of the individual elements: *g_ij_*=cijA/cijU, with each pair of neurons (*i, j*) receiving an individualized attentional modulation *gij*of their shared variability (Figure 2, left). […] To test whether *AC* is low rank we analyzed the V4 population recordings during the visual attention task (Figure 1), specifically measuring *AC* under the assumption that *AC* is rank 1:

**C***A* = **gg***T ◦*
**C***U.”*

*3) Fisher information: contrast versus orientation.*

My main concern is that I see a disconnect between the modeling and the data/experimental paradigm that motivate the modeling.

*I am not convinced about the generality of the conclusions on the effects of attentional modulation on population coding in the model. The experiment is about orientation-change detection, but in the modeling the stimulus dependence is more like contrast (all neurons are identically modulated by the stimulus intensity) than like orientation (where a change in stimulus value would drive some neurons up, and others down). This is acknowledged in the closing paragraphs, and suggested as future work, but I wonder if it should instead be done as part of this paper. I am no expert in EI networks, I don't know how long it would take, but here is concretely what I would like to see and why. Add the stimulus drive in the actual network, not just in the mean field solutions. And while doing that, assume heterogeneity (and possibly nonlinearity) of tuning. If the stimulus acts like contrast, then doing the information analysis on the mean field is fine; but otherwise the mean field solution is effectively a suboptimal decoder (weight all neurons equally), and the conclusions about information may be only valid for that decoder, not for the optimal decoder. The rank-one external noise by itself does not limit information for orientation-like stimulus dimensions (unless you modify it to exactly align it with the signal) (e.g. Moreno-Bote et al., 2014), so some other source of differential correlations needs to be considered if you want the attentional modulation to have any chance of improving information.*

*We'll admit that this may be a hard one to address rigorously. But the authors should provide an extended discussion of this issue. And an attempt should be made to provide approximate calculations and/or estimates.*

We agree with the reviewers that we only grazed the surface of the implications of our attentional modulation mechanism for neural coding. Extending our model to include distributed tuning would be a natural extension, yet one that would be overly cumbersome for this study. In particular, this would involve putting the network on a ring (at minimum) where orientation preference is coded. We would need to choose (and explore) how noise correlations depends on the spatial scale of recurrent interactions, the spatial scale of feedforward stimulus tuning, and even allow attentional modulation to be localized on the ring (to model feature attention). These aspects make a full study worthy of its own report, and we (Doiron and Cohen) currently have a student working on this project.

Rather, we chose to highlight only the simplest consequences of the attentional modulation mechanism we present. If we use a linear framework to study the input-output transfer of both signal and noise then in our simplified model the attentional modulation has no affect on information transfer as decoded from the whole network. However, if the decoder has access to only the excitatory population then 1) the code is suboptimal, and 2) attention can now improve the code.

By virtue of the scalar nature of the model the signal and noise are ‘aligned’ in a trivial sense (*µ* = *ks* + *σξ*(*t*)). The reviewers are correct to point out that should we consider a true population model with distributed tuning where these fluctuations are orthogonal (meaning ‘not parallel’ when speaking about high dimensions) to the dimension over which stimulus is coded then our treatment is overly simplistic. Since extending the model to include a feature dimension is the focus of a later study we will rather interpret our stimulus or population in simpler terms. When first setting up the stimulus we now write:

“Our model captures a bulk firing rate *rE* rather than a population model with distributed tuning. Because of this the stimulus *s* should either be conceived as the contrast of an input, or a population conceived as a collection of identically-tuned neurons (i.e a single cortical column).”

In the section where we analyze the stimulus estimation by our model we now write:

“As mentioned above our simplified mean field model (Equation 6) considers only a bulk response, where any individual neuron tuning is lost. As such a proper analysis of population coding is not possible. Nonetheless, our model has two basic features often associated with enhanced coding, decreased population variability (Figure 5) and increased stimulus-response gain (Figure 7).”

Finally, in the Discussion section we now write:

“Determining the impact of population-wide spiking variability on neural coding is complicated (Averbeck et al., 2006, Kohn et al., 2016). […] It is not clear how noise correlations will depend on these choices yet work in spatially distributed balanced networks shows that solutions can be complex (Rosenbaum et al., 2017).”